# Blockchain-Based Continued Integrity Service for IoT Big Data Management: A Comprehensive Design

**Yustus Eko Oktian**, **Sang-Gon Lee \*** and **Byung-Gook Lee**

College of Software Convergence, Dongseo University, Busan 47011, Korea; yustus.oktian@gmail.com (Y.E.O.); lbg@dongseo.ac.kr (B.-G.L.)
\* Correspondence: nok60@dongseo.ac.kr

**Abstract:** The state-of-the-art centralized Internet of Things (IoT) data flow pipeline has started aging since it cannot cope with the vast number of newly connected IoT devices. As a result, the community begins the transition to a decentralized pipeline to encourage data and resource sharing. However, the move is not trivial. With many instances allocating data or service arbitrarily, how can we guarantee the correctness of IoT data or processes that other parties offer. Furthermore, in case of dispute, how can the IoT data assist in determining which party is guilty of faulty behavior. Finally, the number of Service Level Agreement (SLA) increases as the number of sharing grows. The problem then becomes how we can provide a natural SLA generation and verification that we can automate instead of going through a manual and tedious legalization process through a trusted third party. In this paper, we explore blockchain solutions to answer those issues and propose continued data integrity services for IoT big data management. Specifically, we design five integrity protocols across three phases of IoT operations—during the transmission of IoT data (data in transit), when we physically store the data in the database (data at rest), and at the time of data processing (data in process). In each phase, we first lay out our motivations and survey the related blockchain solutions from the literature. We then use curated papers from our surveys as building blocks in designing the protocol. Using our proposal, we augment the overall value of IoT data and commands, generated in the IoT system, as they are now tamper-proof, verifiable, non-repudiable, and more robust.

**Keywords:** blockchain; IoT; integrity service; data in transit; data at rest; data in process

## 1. Introduction

Since Kevin Ashton coined the term Internet of Things (IoT) [1], engineers have utilized centralized IoT architecture for years. Many state-of-the-art IoT products are still adopting this approach, such as Philips Hue and Amazon Alexa [2]. Several factors lead to this centralization. First of all, the IoT economics forces IoT devices to become constrained-resource devices in favor of low prices to encourage many adoptions. The company then builds central remote servers that gather IoT data and overtake those devices' required computations. Eventually, the server piles up IoT data and provides better IoT services that, in the long run, may be worth more than the company's previous investment on those IoT devices' hardware. Furthermore, IoT big data management is easy to manage in a centralized architecture. In terms of security, the administrations only need to secure the servers where the IoT data resides and sometimes, ironically, neglect IoT devices' security. They can also provide data sharing for other parties through Application Programming Interfaces (APIs). Since the company can do multiple things from a single place (i.e., the servers in the Cloud), it is convenient.

As IoT applications become sophisticated, centralized architecture begins to suffer from a single point of failure, poor scalability, high latency, and privacy issues. To overcome these issues, the community begins the transition to a decentralized IoT architecture (cf. [3] for detailed comparisons

of centralized and decentralized IoT architecture). However, this movement is challenging to realize. The IoT data flow now becomes more complicated because many entities share data, commands, and resources. This complexity results in more variations to the IoT data flow pipeline and augments data fragmentation. Poor management and unauthorized tampering in one part of the pipeline may result in inefficiency or malfunction on other IoT operations, causing economic loss. Therefore, we need a reliable ecosystem that can log arbitrary IoT data flows to guarantee the correctness of the IoT process in this new decentralized architecture.

The rise of blockchain, which comes with the popularity of Bitcoin [4], exhibits a promising potential of decentralization through its cryptocurrency. Many researchers then argue that they can also apply blockchain to other non-cryptocurrency domains. For example, in IoT, blockchain can facilitate the sharing of data and resources, create a marketplace for IoT entities, and allow automation in some parts of the IoT processes with verifiable property [5].

In synergy with those researchers' claims, this paper investigates whether the blockchain can provide a continued integrity service for IoT big data management. Our discussion focuses on integrity issues of the IoT data flow pipeline across three IoT operation phases—during the transmission of IoT data and commands (data in transit), when we store them in the database (data at rest), and at processing time (data in process). We first conduct a literature survey on the state-of-the-art blockchain projects related to each of the phases and find several blockchain-based strategies. However, their proposed solutions tackle specific issues. Meanwhile, we argue that the decentralized IoT processes are interdependent on one another. Therefore, a grand design is required to provide a continual integrity service throughout those phases.

We fill the research gap by proposing five integrity protocol designs—decentralized identity management, secure channel establishment, blockchain receipts with the chain of signatures, decentralized e-marketplace, and collaborative federated learning. We intertwine all of those proposals to create a continued integrity service throughout IoT big data management. Our initial surveys also play a role as building blocks for our designs. Using our proposed model, we augment the overall value of IoT data and commands generated in the IoT system as they are now tamper-proof, verifiable, non-repudiable, and more robust.

In summary, we made the following contributions:

- We propose blockchain-based integrity services for data in transit by proposing decentralized identity management and secure channel establishment. Our designs comprise registration, update, and revocation of the public key and the associated domain names. We also present a secure and reliable Transport Layer Security (TLS) or Datagram Transport Layer Security (DTLS) secure channel using our identity management as its foundation.
- We investigate using the chain of signatures and blockchain receipts to provide data integrity during data at rest for IoT databases. Using both combinations, we can preserve the forensically sound guarantee of the stored raw IoT data and commands.
- We leverage the blockchain to empower the decentralized marketplace and federated learning to encourage IoT entities' collaborations. This proposal augments the overall data's robustness because blockchain logs every action in IoT processes. As a result, entities can offer IoT data training services or participate in federated learning processes in a fair, transparent, secure, and verifiable manner.

We organize the rest of this paper as follows. We revisit some backgrounds about IoT big data management in Section 2. Section 3 describes our proposed design to provide data integrity for IoT data transmissions. Section 4 lays out our approach using the chain of signatures and blockchain receipts to offer a data integrity service for stored IoT raw data and commands. Section 5 elaborates on our blockchain-based marketplace and federated learning to facilitate the data training collaboration among IoT entities. Afterward, we discuss how our proposed design can solve open problems related to IoT big data management as well as future considerations and challenges in Section 6. Finally, we conclude in Section 7.

## 2. IoT Big Data Management

In this section, we explore what the IoT data flow pipeline looks like in the IoT big data environment. Then, we describe typical IoT data flow pipeline options that developers can apply in their IoT systems. Finally, we discuss the IoT data flow pipeline's common issues and challenges and present blockchain technology contributions in solving those mentioned obstacles.

### 2.1. Characteristics of IoT Data Flow Pipeline

Raw IoT data from a single device tell us little information, like knowing our neighborhood's current temperature. It can be useful to some extent; however, it can be more appealing if we have more data or combine it with other IoT devices. For example, by gathering the sensor's temperature reading periodically (e.g., every 10 min) for a day (24 h, from 12 AM to 12 PM), we can determine the high and low temperatures for that particular day. It also enables us to detect potential global warming if we track and store temperature data for decades (by comparing high and low data yearly). Moreover, installing additional temperature sensors scattered across multiple neighborhoods gives us more precious insights, such as understanding the average temperature for overall neighborhoods or determining the hottest or coldest areas.

Unfortunately, IoT devices are mostly machines that have constrained resources. Thus, they cannot process IoT data by themselves. The state-of-the-art IoT data process is to do it in powerful servers far away from the devices. We commonly define this location as the Cloud. This method forms a widely known IoT data flow pipeline, as illustrated in Figure 1a. We can categorize this data flow into five layers with three operations types, which we describe as follows.

The sensing layer comprises IoT devices, which can be sensors or actuators. Sensors generate IoT data for the IoT system, and actuators listen for commands from IoT services. In the middleware layer, IoT gateways sit between IoT devices and services to intercept IoT data from the devices and conduct microprocessing. It can be a data clean-up process, where the gateway removes duplicates or invalid data before sending them to IoT services, or it can also be the aggregation of data, where the gateway groups similar data (e.g., using minimum, maximum, or average summarization) into one IoT data. IoT services reside in the application layer to store aggregated IoT data and to serve it to IoT users. To gain insights, IoT workers dwell in the processing layer to train the IoT data using machine learning algorithms. The workers send trained analytic results back to IoT services, which the services then may transfer to users or devices as feedbacks. Finally, the network layer is responsible for the delivery of data between IoT entities. The channel between devices and gateways are varied, and we can use many options such as Bluetooth, Zigbee, and 6LowPAN. Meanwhile, we can use coherent well-known TCP/IP stacks to deliver messages from gateways to services, services to workers, and vice versa.

Throughout those previously mentioned layers, we have three types of IoT operations. First, data in transit includes procedures that we take during transmissions of IoT data. These steps mostly happen in the network layer, where entities pass the IoT data from one to another. Second, data at rest is a process where we store IoT data in the local database. The middleware layer may store IoT data temporarily for real-time application use cases. They will delete this data when it is no longer needed. Meanwhile, the application layer saves IoT data permanently in the database for analytic and presentation purposes. This layer is capable of storing many data, usually clustered in several servers. Third, we process the IoT data in a stage that we called data in process. These operations may include micro- and macro-processings, which happen in the middleware and processing layers.

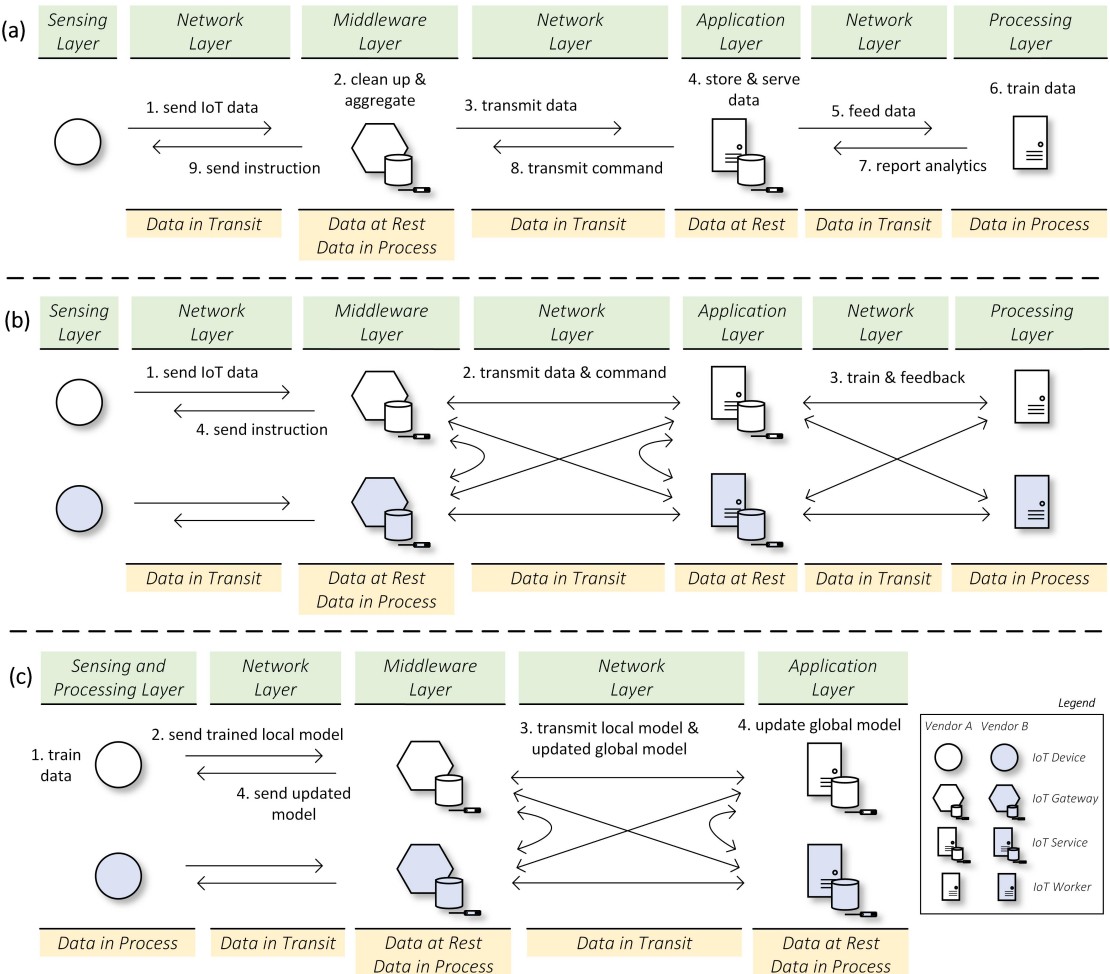

**Figure 1.** A general Internet of Things (IoT) data flow pipeline, which comprises of 5 layers (i.e., sensing, network, middleware, application, and processing layers) and 3 types of operations (i.e., data in transit, data at rest, and data in process). We present three IoT data flow pipeline scenarios: (**a**) the centralized IoT data flow pipeline, (**b**) the collaborative IoT data flow pipeline, and (**c**) the privacy-preserving IoT data flow pipeline.

## 2.2. Types of IoT Data Flow Pipeline

**Centralized IoT Data Flow Pipeline**: in this first category, all of the required IoT data are gathered and stored centrally in IoT services. Therefore, management (in terms of security, data sharing, and service provisioning) becomes easy. In early IoT systems, when a particular company wants to develop a new IoT platform, they build their framework (from IoT devices to IoT workers) following this data flow pipeline as a template, shown in Figure 1a. As a result, this development encourages many siloed architectures to exist among corporations, creating inefficiency and disorganization in IoT pipelines with many implementation variations. Moreover, the nature of this architecture allows vendors to interpret IoT devices as slaves because most of them are passive devices that collect data for IoT services. Therefore, this phenomenon creates ethical malpractices. For example, developers usually put more focus on how they can provide better services using all of the already collected data and neglect the condition of the IoT devices' hardware, software, or both by rarely providing new features or security updates. Last but not least, this centralized pipeline is vulnerable to single-point-of-failure and scalability issues. The increased latency due to Cloud processing also hinders the applicability of real-time processing. Therefore, we argue that this architecture is more suitable for pilot programs or small-scale IoT operations.

**Collaborative IoT Data Flow Pipeline**: one key idea to alleviate the weakness of the centralized pipeline is through data sharing and collaboration. As shown in Figure 1b, these actions can be performed in several layers. The vendor can move some IoT data processing from the application layer to the middleware layer. Thus, IoT gateways can do more complicated operations besides cleaning up and aggregations by acting as the secondhand of the IoT services. We usually coin this process as fog computing [6]. By processing data in IoT gateways, IoT events are now faster to complete. We can then realize the IoT real-time application use cases. Moreover, the gateways can also act as temporary machines to serve IoT users' requests when the actual IoT services are crashing or under heavy load, increasing the system's scalability. IoT services can mutually share the collected IoT data with other services to provide better analytics in the application layer. Finally, in the processing layer, the IoT vendor can let other vendors use its IoT workers' resources to train other vendors' data on their behalf. The vendor can do so when its workers are idle, thus producing additional revenue for the vendor.

**Privacy-Preserving IoT Data Flow Pipeline**: while the collaborative approach helps in reducing the latency and increasing overall scalability, it still does not solve data privacy issues. In particular, IoT data still needs to be sent to IoT services to train it. If this data is very sensitive, then IoT users are at a disadvantage because there is no other way to benefit from the IoT system's analytics without giving up their private data. Therefore, a recent trend arises regarding a fully distributed privacy-preserving IoT data flow pipeline, depicted in Figure 1c. The similarity between this pipeline and the collaborative one is that we can still find cooperation between IoT gateways and IoT services. Meanwhile, the difference is that we remove IoT workers from the architecture. IoT devices will instead train their private data in their local machine and share only local trained models to the IoT gateways and services. In this sense, IoT devices become both sensing and processing layers. This approach preserves users' privacy since the gateways and services only understand the local models and cannot view private data.

*2.3. Basic Challenges and Requirements for IoT Data Flow Pipeline*

Those three previously mentioned pipelines have their merit and disadvantages. Developers can choose to deploy one of those pipelines according to their needs. Regardless of their choices, some common challenges and requirements are still required to ensure the IoT data flow pipeline's integrity. We analyze them, along with the corresponding blockchain solutions in Table 1, and describe them as follows.

**Identity**: all entities that participate in the IoT process must have a unique identity. Otherwise, we cannot conduct proper data management and determine an IoT data or process's origin. For IoT devices, the identity can be the Universal Unique Identifier (UUID) [7], a global identifiable ID that the IoT manufacturers embedded in IoT devices. For IoT services or users, the identity can be a combination of usernames and passwords. Both UUID and username require central server identity management, which is prone to credential theft [8]. On the other hand, blockchain identifies users in the network by their unique addresses, derived from public keys. Public key generation is decentralized, and any node can create a new public key without contacting a centralized third party. Thus, we can use the blockchain public key as a global identity system in our architecture.

**Non-repudiation**: IoT data and processes may come from anywhere. Attackers can pretend to be one of the authorized entities and can send malicious data or processes. In these circumstances, the IoT pipeline should detect attackers and ensure that they cannot deny or make valid excuses to revoke the fact that they transmitted malicious detected packets. When users store information as transactions in the blockchain, they must first sign the transactions before broadcasting them to other peers. The signatures serve as proof that the users know the transactions. Anyone can verify the authenticity of the signature, and the users cannot deny their own signature. Moreover, this feature is available not only in the blockchain network (on-chain) but also in the non-blockchain network (off-chain), e.g., the Internet or IoT network. Therefore, we can also use the same blockchain public key architecture to sign arbitrary IoT data and processes off-chain.

**Tamper-proof**: the IoT environment requires having tamper-proof data storage and process logs, in which attackers should not be able to modify the contents. Malicious changes can generate malfunctions, inaccurate predictions, or both, that may result in economic loss. The blockchain aggregates transactions into blocks. Each block has its own hash that will protect its integrity. Furthermore, the blockchain requires each block's hash to be referenced to the next block extending it. Therefore, the longer the blockchain extends a particular block, the more difficult it is to tamper its contents. Any modification in that block requires the attackers to modify the next blocks' contents to be considered a valid chain of blocks. Finally, the data stored in the blockchain is also fully distributed. Attackers then need to modify the contents in the majority of the blockchain nodes in the network to successfully tamper a particular data, which is a challenging task to achieve.

**Fault-tolerant**: the IoT process should be able to continue working even during crashes or failures. In the IoT system, failures in one part of the data flow pipeline can be catastrophic as it may hinder progress in other parts of the pipeline. In the blockchain, all nodes store the data ledger locally and synchronize it with other nodes. This concept is similar to a replicated state machine in the distributed system. This mechanism guarantees decentralized control with no single point of failure, increasing the IoT system's robustness.

**Confidentiality**: with many entities sharing data and resources in the IoT pipeline, there can be cases when vendors or users do not want other entities to know their contents, especially when the data is very private and sensitive [9]. Thus, it is useful if the system has options to protect the secrecy of these private data and resources by using encryption. Unfortunately, the data stored in the conventional blockchain, such as in Bitcoin [4] or Ethereum [10] are not encrypted. However, using blockchain public keys as building blocks, we can construct secure encryption schemes. For example, Quorum [11] modifies the Ethereum blockchain to enable private transactions, which encrypts transactions to be understood only by selective receivers. There is also an Ethereum library [12] to encrypt arbitrary data off-chain using Ethereum public keys.

**Privacy**: IoT vendors mostly conduct a massive data collection strategy to produce useful insights for their services. They often perform this operation by sacrificing users' privacy [13], threatening the trust and usability of overall IoT systems. Blockchain cannot fully solve this problem; however, it can help users achieve pseudonymity. By using blockchain public keys, users can generate arbitrary identities each time they contact the IoT systems. Attackers will find it very difficult to relate one public key to another and pinpoint the real users of the public keys' holders.

**Trusted SLA**: the primary issue that hinders collaborations between IoT vendors is trust. By default, one vendor sees others as competitors; therefore, they act with complete distrust. When a vendor wants to perform partnerships with others, both parties need to create a Service Level Agreement (SLA) that will act as a legal document that protects their agreement. State-of-the-art SLA generations require yet another trusted third party. Moreover, it also takes time to produce such SLAs as it may include manual labor processes. Blockchain can help to create automatic trusted SLAs between parties in the form of smart contracts. The smart contracts' code is not only deterministic but also open in the blockchain. Anyone can verify the smart contract's source code, so others can safely trust smart contracts' execution.

To sum up, we argue that blockchain is a suitable platform candidate to provide integrity services for IoT big data management. In the following sections, we lay out how we can use blockchain to protect the integrity of the IoT data flow during three phases of IoT operations: data in transit, data at rest, and data in process. In each part, we first present our motivations for the discussed phase. Then, we introduce our proposed design and analyze its benefits.

**Table 1.** Basic requirements for the management of the IoT data flow pipeline.

| Properties | Challenges | Blockchain Solutions |
|---|---|---|
| Identity | Given an IoT data or process, we need to determine who the source of that data or process is. | Blockchain provides public keys that we can use as verifiable identities. |
| Non-repudiation | The origin of IoT data or processes should not deny that they generate such data or processes. | For the sender to store information in the blockchain, they must sign the data as transactions. |
| Tamper-proof | Attackers should not be able to modify the contents of IoT data or process logs maliciously. | Blockchain stages the data into a block data structure with the hash of the block chained to the previous block to produce a hard-to-tamper reference. |
| Fault-tolerant | The data management should be functional despite failures that may happen during its lifetime. | Blockchain network is fully distributed; therefore, it eliminates the single point of failure and increases the system's robustness. |
| Confidentiality | The system requires having the capability to encrypt sensitive information from being leaked to the public. | Blockchain does not provide this ability. However, we can use the public keys that the blockchain has to provide secure encryption. |
| Privacy | We need to protect users' privacy by protecting their private data when conducting massive data collections and processings. | The use of public keys in the blockchain renders pseudonymity and strengthens users' privacy. |
| Trusted SLA | Trust issues hinder collaboration, and producing SLAs is lengthy and tedious task. | Using smart contracts, we can automate the trusted SLA generation process. |

## 3. Blockchain Solutions for Data in Transit

This section investigates integrity services for IoT data and commands that IoT entities transmit in the system. More specifically, we focus on identity management and deploying a secure channel to protect data transmission.

### 3.1. Motivations

To protect the integrity of the transmitted IoT data through the Internet, we commonly construct a secure channel, a transmission medium that covers several security guarantees. First, the channel assures that participants are communicating with the right entities, not fake ones. Second, only those involved parties can see the delivered data. Others should not be able to gain any knowledge about the content of transmitted information. Finally, the members are confident that the data they receive is original such that no other entities tamper with the data.

We can use existing encryption and digital signature schemes to build a secure channel. However, we find that the underlying components of the secure channel, mainly identity service providers such as Public Key Infrastructure (PKI) [14] and Domain Name Service (DNS) [15], pose centralization threats that may hinder collaborations between entities in our IoT data flow pipeline.

In PKI architecture, a centralized Certificate Authority (CA) exists as a trusted third party. Other entities can let CAs sign their public key to enforce trust in them. If outsiders can prove that the CA signed a particular public key, they can safely assume that it belongs to a valid entity because the CA already verified it. In other words, they accept and trust any public key that the CA signs.

Because of the absolute power that the CA has, if the CA misbehaves (e.g., due to attacker infiltration or misconfigurations), the damage can be catastrophic [16]. The CA can flag a hazardous website that contains malware as valid and safe. Users then visit the site and are infected by the

virus. Furthermore, we also argue that the current PKI solutions cannot cope with the scale of IoT devices. Constrained Application Protocol (CoAP) [17] and Message Queuing Telemetry Transport (MQTT) protocol [18] allow IoT users or services to query IoT data to IoT devices directly through gateways. IoT gateways now behave like servers and need to serve many requests from users and services. Therefore, the CA now has additional jobs to sign public keys from many gateways.

Similar issues occur in the DNS as well. The commonly used DNS system stores the mapping between domain names and IP addresses in a hierarchical and centralized manner. Thus, DNS servers, especially DNS root servers, have absolute authority over the registered names. Using this DNS architecture, it raises the same problems as we previously stated in the PKI system.

To solve the previously mentioned issues, we have no other options but to reduce or eliminate centralization. We can use blockchain as a platform for decentralization, as proposed in [19] for PKI and in [20] for DNS. However, to the best of our knowledge, no comprehensive design shows how those decentralized identities can be used to generate a secure channel. To fill that gap, we design both decentralized identity management and secure channel establishment using blockchain. While plotting our design, we also carry out a literature survey on other blockchain-related solutions regarding data in transit. We then apply the chosen papers from our survey as our design's building blocks. We summarize their contributions to our design in Table 2.

**Table 2.** The list of related works serving as building blocks for our data in transit solutions.

| Project | Related to | Contributions to Our Design |
|---|---|---|
| Yakubov et al. [21] | Identity Management | We apply the proposed idea to extend X.509 certificate fields for our blockchain integration in a quasi-centralized approach. |
| SCPKI [19] | Identity Management | We use the presented WoT scheme to produce a certificate in our fully decentralized method. |
| Wilson and Ateniese [22] | Identity Management | We take the authors' suggestions regarding the incentive mechanism for public key endorsements in our design. |
| CertCoin [23] | Domain Names, Identity Management | The authors introduce two-key management: online and offline key, which we tweak it to work with our smart contract. |
| Kalodner et al. [24] | Domain Names | We acknowledge the authors' namespace business model recommendations for our future design. |
| BATM [25] | Reputation | We assume using the proposed reputation system in our design, which considers positive or negative reviews and their timestamp. |
| PADVA [26] | Secure Channel, Reputation | We apply the proposed TLS timestamping in our design, which makes use of `client-random` and `server-random` to prove that TLS handshakes are performed. |

### 3.2. Our Proposed Solutions

We divide our explanations into two segments: decentralized identity management and secure channel establishment. They are all parts of the overall IoT data in transit integrity. We present new notations that we employ throughout the rest of this paper in Table 3.

**Table 3.** The list of new notations introduced in our data in transit design.

| Notation | Description |
|---|---|
| $X$ | The IoT entity, which can be an IoT device, gateway, service, or worker. They are defined as $D$, $GW$, $S$, and $W$, respectively. |
| $PK_X^{online}$ | The online public key of $X$ entity. |
| $SK_X^{online}$ | The online secret key of $X$ entity. |
| $\alpha_X^{online}$ | The online blockchain address of $X$ entity. |
| $PK_X^{offline}$ | The offline public key of $X$ entity. |
| $SK_X^{offline}$ | The offline secret key of $X$ entity. |
| $\alpha_X^{offline}$ | The offline blockcahin address of $X$ entity. |
| $d_X$ | The arbitrary domain name of $X$ entity. |
| $\gamma_X$ | The public IP address of $X$ entity. |
| $SIGN_{SK_X}(J)$ | A function to generate a signature using any asymmetric digital signature algorithm (e.g., ECDSA) from payload $J$ using the secret key of $X$ entity, $SK_X$. |
| $sig$ | The generated signature output of the $SIGN(.)$ function. |
| $cert$ | The certificate as proof of identity (e.g., X.509 certificate). |
| $random$ | The `client-random` or `server-random` parameter from TLS handshake. |
| $VERIFY_{PK_X}(J, sig)$ | A function to verify whether the signature of $J$ payload, $sig$, is produced using a secret key corresponding to $PK_X$ public key. This function returns `True` or `False`. |
| $A \parallel B$ | A concatenation between $A$ data and $B$ data. |

### 3.2.1. Decentralized Identity Management

Our decentralized identity management design becomes the foundation for all IoT entities to recognize and verify one another. We will use this identity scheme throughout the rest of our proposals. Figure 2 summarizes our design.

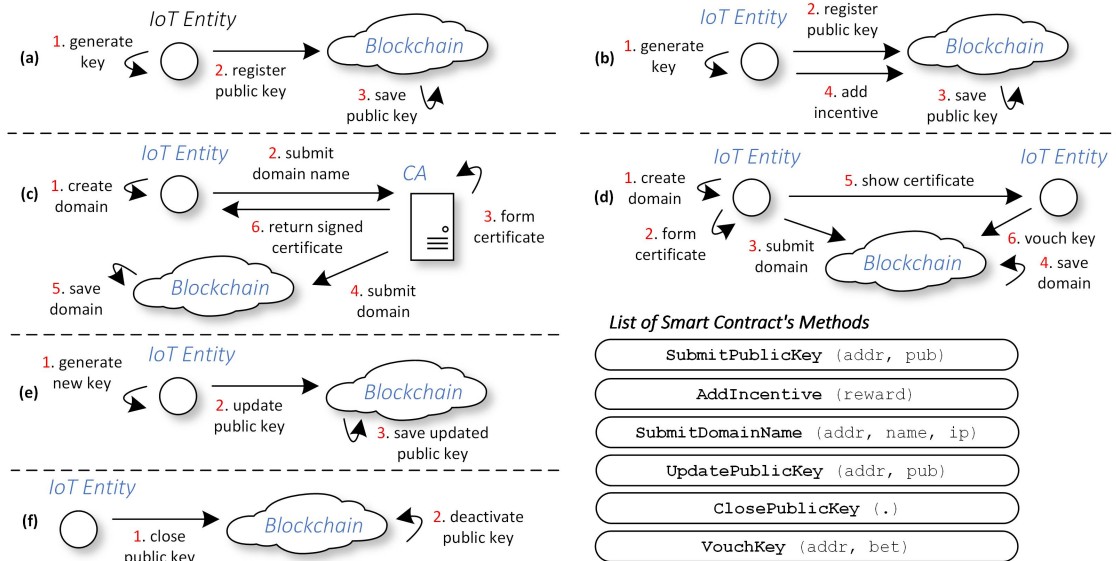

**Figure 2.** The overview of our decentralized identity management design: It comprises of public key registration for (**a**) the quasi-centralized and (**b**) fully decentralized approaches, namespace registration for (**c**) the quasi-centralized and (**d**) fully decentralized methods, (**e**) online key revocation, and (**f**) offline key revocation. We also show the list of required smart contract methods for implementation.

**Decentralization Options**: we can choose whether to make our system becomes quasi-centralized or fully decentralized.

- Quasi-centralized approach reduces CA and the DNS server's centralization impact by implementing a kind of certificate transparency [27] in the blockchain. When we store a list

of identities in the blockchain, the list becomes open to all nodes. Thus, they can verify an IoT entity's trustworthiness and can expose the possibility to detect malicious identity or misbehaving CA and DNS servers quickly.

- The fully decentralized method removes the role of the CA or DNS server altogether, and all participants collectively manage the identity management. Specifically, we employ the Pretty Good Privacy (PGP) technique [28], where we manage identity distributedly in the form of Web of Trust (WoT) [29]. In this second option, we leverage blockchain as a platform to perform WoT.

IoT users, gateways, services, notaries, or any party that wants their name to be identifiable by others can begin registering their identity to the blockchain on the "public bulletin board". They need to register both their public key and specified namespace. Our design is inspired by CertCoin [23], which provides an excellent key management example. We take the general idea from that proposal and tweak it to be compatible with our smart contract design.

**Public Key Registration**: the following steps describe our public key registration procedure.

1. The entity generates two pairs of keys—the online and offline key—using ECDSA key-generation procedure similar to the one used to generate the Ethereum address [30]. $PK_X^{online}$, $SK_X^{online}$, $PK_X^{offline}$, and $SK_X^{offline}$ are the online public key, online secret key, offline public key, and offline secret key, respectively. From these keys, the entity also generates the corresponding online and offline addresses $\alpha_X^{online}$ and $\alpha_X^{offline}$. $X$ refers to the entity's role in the IoT system, whether it is an IoT device, gateway, service, or worker. Thus, $X = \{D, GW, S, W\}$.
2. The entity submits $\alpha_X^{online}$ and $PK_X^{online}$ to the smart contract by calling the `SubmitPublicKey(addr, pub)` method. The `addr` and `pub` are the entity's online address and online public key to be submitted. The entity must use its offline address, $\alpha_X^{offline}$, as the sender when forming the transaction for this method.
3. The smart contract maintains the mapping between offline and online keys in key-value storage. Upon receiving the transaction from the previous step, the smart contract retrieves $\alpha_X^{online}$ and $PK_X^{online}$. It also queries $\alpha_X^{offline}$ from the sender parameter. The smart contract then saves a new entry in the list using $\alpha_X^{offline}$ as keys and $\{\alpha_X^{online} \parallel PK_X^{online}\}$ as its values.
4. In the fully decentralized approach, the entity calls the `AddIncentive(reward)` method to add a prize to encourage other entities to vouch for this key. The `reward` is the award that endorsers of this public key can claim later. The entity also must use their offline address, $\alpha_X^{offline}$, as the sender when creating the transaction for this method. We take this idea from Wilson and Ateniese [22], which discuss the endorsement issues of PGP and WoT for a newly registered key.

The rationale in using two keys, online and offline keys, is for safety reasons. In particular, there is a possibility that attackers can guess the secret key, especially if we do not use a secure seed during key generation [31]. By having a secondary offline key as a backup, we can overrule our online key if attackers correctly guess or steal it. This security is guaranteed for two reasons. First, we use the online key mainly for off-chain use cases (i.e., things unrelated to the blockchain network). Meanwhile, the offline key is used only for on-chain use cases (i.e., blockchain network-related). On-chain scenarios happen less frequently than off-chain ones. Thus, it minimizes the leak probability of our offline key. Second, we assume that the entity put more security precautions on the offline key than the online key. For example, the entity puts the offline secret key in a separate and more secure machine than the online secret key.

**Namespace Registration**: after the public key is registered, we can continue to record our namespace. The smart contract will tie this namespace to the already submitted online public key. The entity can make arbitrary namespace, and the system can support many categories. However, to simplify our explanation, we only consider the namespace of domain names for DNS service in this paper.

For the quasi-centralized approach, we do the following.

1. The entity creates an arbitrary domain name, $d_X$. It also retrieves the public IP, $\gamma_X$, from the Internet Service Provider (ISP).
2. The entity submits $d_X$ and $\gamma_X$ along with $\alpha_X^{offline}$, $\alpha_X^{online}$, and $PK_X^{online}$ to the CA off-chain. It also needs to provide detailed information regarding proof of domain name and public IP address possession for legal purposes. The CA verifies the proof and makes sure that the submitted domain name is unique.
3. The CA then forms a certificate that testifies the entity's possession of the domain name. In particular, the CA creates X.509 certificates for TLS or DTLS scenarios. We borrow the X.509 extensions from [21]. The authors suggest to include blockchain information such as the address of the smart contract, the CA's blockchain address, and the hashing algorithm as additional info in X.509 fields. Therefore, when another party receives this signature, that party can pinpoint the issuers' details and determine which smart contract they contact to verify the signature.

   The CA includes $\alpha_X^{online}$, $PK_X^{online}$, and additional info suggested from [21] in X.509 fields. It then signs the certificate using the CA's secret key, $sig_{CA} = SIGN_{SK_{CA}^{online}}(cert_X)$. $cert_X$ refers to the X.509 certificate for $X$ entity. Finally, the CA includes its signature in the certificate, $cert_X \leftarrow sig_{CA} \parallel cert_X$.

4. The CA submits the $\alpha_X^{offline}$, $d_X$, and $\gamma_X$ to the smart contract by calling the `SubmitDomainName(addr, name, ip)` method. `addr`, `name`, and `ip` are the offline address, domain, and the public IP address name to be submitted, respectively.

   For simplicity, in this example, the domain name registration is free. Kalodner et al. [24] conducted an extensive empirical analysis of Namecoin [20], a working example of a decentralized domain name system run in the blockchain. Their research concluded that Namecoin is in poor condition, mainly because of economic reasons. To mitigate similar failures, the authors provide insights regarding a better-decentralized namespace design by allowing administrators to choose five options of controls from the most robust control to the weakest one. They also suggested using auctions, algorithmic pricing, and secondary market to boost decentralized domain names' economy. We can take the authors' suggestion when designing our future economic model.

5. The smart contract maintains a list of registered domain names in a key-value store. Upon receiving the transaction in the previous step, the smart contract puts $d_X$ as the key and $\{\alpha_X^{offline} \parallel \gamma_X\}$ as the value in the store. We apply domain names as keys to ensure the uniqueness of the stored namespaces. We save $\gamma_X$ to map the domain name with its IP address. The role of $\alpha_X^{offline}$ as a value is to bridge this domain name list and the mapping of offline and online keys storage. Therefore, from this link, the smart contract can find a relationship between $d_X$ and its associated online identities, $\alpha_X^{online}$, and $PK_X^{online}$.
6. The CA returns $cert_X$ to the entity. The entity can then present this certificate during the handshake of the secure channel establishment that we explain in the next section. By default, other users will trust this certificate because the CA signed it.

   For the fully decentralized approach, we do the following.

1. The entity creates an arbitrary domain name, $d_X$. They also receive the public IP, $\gamma_X$, from the ISP.
2. Because there is no CA in this approach, the entity forms a certificate by itself, which produces proof regarding possession of the domain. We borrow the certificate format proposed in SCPKI [19]. Some mandatory information includes the smart contract address, the entity's online public key, and its online address. These parameters are essential to guide other users in the verification of the certificate.

   After that, the entitiy signs the certificate with its online secret key, $sig_X = SIGN_{SK_X^{online}}(cert_X)$. $cert_X$ refers to $X$'s certificate. The entity then includes its signature in the certificate, $cert_X \leftarrow sig_X \parallel cert_X$.

3.  The entity submits $\alpha_X^{offline}$, $d_X$, and $\gamma_X$ to the smart contract by calling the `SubmitDomainName(addr, name, ip)` method.
4.  Similar to the quasi-centralized approach, the smart contract then puts $d_X$ as the key and $\{\alpha_X^{offline} \parallel \gamma_X\}$ as the value in the storage.
5.  Once it is stored, the entity can present $cert_X$ during the handshake of secure channel establishment.

By default, other users will not trust the entity's certificate because it is a self-signed certificate. Similar to the methodology in the WoT, to build trust upon this certificate, other users need to vouch for or approve of this certificate by signing it using their online secret key.

6.  To endorse a particular certificate, IoT entities can vouch for the certificate's corresponding online address. The IoT entity calls the `VouchKey(addr, bet)` method. `addr` and `bet` are the online address and the stake to endorse the given address.

We draw the betting idea from Wilson and Ateniese [22], which suggests that users specify the amount of money they are willing to risk to verify a particular public key. The system adjusts the trust level by considering the number of deposits, with a higher wager equal to higher trust. Moreover, the system also determines the incentive that endorsers can take by considering their initial bet. Similarly, punishment (in the form of reduced reputation) also increases as the wager rises.

During IoT operations, attackers can infiltrate our system and steal our identities. Therefore, we design two revocation procedures to protect our identity system.

**Revoking the online key**: let us say that attackers correctly guess or steal the online secret key. However, possession of the offline secret key is still safe and owned by the entity. The entity can outsmart the attackers by revoking the old online key and updating it with a new online key.

1.  The entity generates a new pair of online key and its address, $SK_X^{online'}$, $PK_X^{online'}$, and $\alpha_X^{online'}$.
2.  The entity then uploads both the key and address to the smart contract by creating a transaction that calls the `UpdatePublicKey(addr, pub)` method. `addr` and `pub` are the new online blockchain address and public key. Note that they need to use the old offline address, $\alpha_X^{offline}$, as the sender of the transaction when calling this method.
3.  Upon receiving this transaction, the smart contract checks the sender and ensures that it equals the previously stored address during identity registration. This check is to guarantee that only the original submitter can update the online keys. If everything is valid, the smart contract stores the new online keys in the database.

Note that the entity must also reconfigure its old certificate to match the newly updated online public key. It can do so by refollowing the namespace registration procedure.

**Revoking the offline key**: let us say that attackers correctly guess the offline secret key. However, the entity still has its offline key. Then, the entity can outsmart the attackers by closing its key.

1.  The entity creates a transaction that calls the `ClosePublicKey(.)` method. They need to use the previously registered offline address, $\alpha_X^{offline}$, as the sender when forming the transaction.
2.  Upon receiving this transaction, the smart contract checks its sender and makes sure it exists in storage. This check is to guarantee that only the original submitter can close online keys. If everything is verified, the smart contract marks the online keys as closed.

The closing procedure is irrevocable. Therefore, once it is closed, the online key and the associated certificate will also become invalid. Furthermore, because we tie $d_X$ with $\alpha_X^{offline}$, the offline key's closing will render the domain names unusable. Unfortunately, the entity cannot transfer this domain name to other offline keys. Therefore, they lose access to their unique domain names. One solution to this issue is to submit yet another backup account for this domain during namespace registration.

Note that we cannot protect the entity if attackers steal the entity's offline secret key, and the entity no longer has access to its key.

### 3.2.2. Secure Channel Establishment

This section shows how we can build a secure channel between IoT entities using our previously described decentralized identity management as its foundation. Figure 3 depicts an overview of our secure channel design.

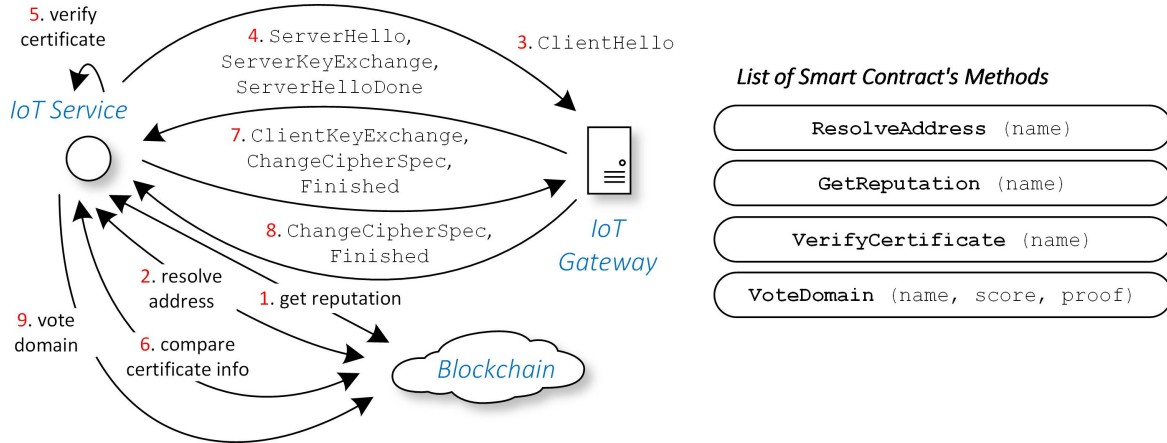

**Figure 3.** Overview of our blockchain-based secure channel design for TLS or DTLS: we also put the list of required smart contract methods for developers to implement.

First of all, we make the following assumptions.

- A trusted party deploys the smart contract in the blockchain network. For example, the CA is responsible for this role in the quasi-centralized approach. For the fully decentralized method, the government can serve this role.
- We employ IoT services and gateways in this scenario merely as an example. One can reuse the protocol to build a secure channel between any IoT entities.
- The gateway already registered its public key, address, namespace, and IP address in the blockchain, specifically $PK_{GW}^{online}$, $\alpha_{GW}^{online}$, $d_{GW}$, and $\gamma_{GW}$. Moreover, the gateway's certificate, $cert_{GW}$, is also ready to use.
- The gateway provides IoT data and service by allowing IoT services to access the IoT domain through a URL (e.g., `https://gateway.bit`) tied to $\gamma_{GW}$. Thus, $d_{GW}$ is `gateway.bit`.
- The IoT gateway already built some positive reputation scores in the system.

We endorse the use of a reputation system proposed in BATM [25]. The authors define five reputation factors, which include negative and positive events. Their proposal also considers the freshness of the events by factoring the formula using a continuously decreasing function. Therefore, the latest events will have more weight than old events, and they will contribute more to the final reputation score.

- The secure channel protocol is based on TLS [32] for TCP traffics or DTLS [33] for UDP traffics.

Then, the IoT service builds a secure channel with the IoT gateway by following these steps.

1. The IoT service is trying to access IoT contents from the URL (e.g., `https://gateway.bit`). Optionally, before the service accesses the URL, it invokes the `GetReputation(name)` method in the smart contract by including the `gateway.bit` (i.e., $d_{GW}$) as the `name` argument. The smart contract then returns the reputation score for the `gateway.bit` domain.
2. Assuming that the IoT service is satisfied with the reputation score, it begins to access the URL. We assume that this is the first time the IoT service accesses this URL. Therefore, the service conducts DNS operations to resolve the IP address for the given domain. It calls the

ResolveAddress(name) method and puts gateway.bit as the name argument. The smart contract returns the domain's IP address, $\gamma_{GW}$.

3. The IoT service begins the initial TLS handshake. During this exchange, the service must gather proof that it participates in the handshake to later vote for the gateway's certificate. For this purpose, we borrow the concept of TLS notary in PADVA [26].

First, as a client, the IoT service begins the handshake by sending a ClientHello message. The service also generates client-random as the client part of the exchange to produce the final ephemeral secret key. This random generation includes the UNIX timestamp, which can serve as our proof.

4. The gateway replies with ServerHello, ServerKeyExchange, and ServerHelloDone messages.

ServerKeyExchange includes the server-random property as the server part of the exchange to build the final ephemeral secret key. The gateway puts a UNIX timestamp in this random, which can serve as our proof. Moreover, the gateway also signs the server-random, defined as $random_{GW}$, and generates the signature, $sig_{random_{GW}} = SIGN_{SK_{GW}^{online}}(random_{GW})$.

Note that the gateway also sends their registered certificates $cert_{GW}$ to the user at this step.

5. Upon receiving the $cert_{GW}$, the IoT service retrieves the gateway's public key, $PK_{GW}^{online}$, from the certificate and verifies that $VERIFY_{PK_{GW}^{online}}(random_{GW}, sig_{random_{GW}})$ returns True. After that, the service validates the rest of the certificate.

For the quasi-centralized approach, the service checks if the CA signs $cert_{GW}$ and $VERIFY_{PK_{CA}^{online}}(cert_{GW}, sig_{CA})$ must return True. Meanwhile, for the fully decentralized method, it verifies that the gateway indeed signs the certificate and that $VERIFY_{PK_{GW}^{online}}(cert_{GW}, sig_{GW})$ must return True.

6. The service compares the required fields in the certificate with the ones stored in the smart contract. It invokes the VerifyCertificate(name) method with gateway.bit as an argument. First, the smart contract makes sure that the certificate (i.e., identifiable by $PK_{GW}^{online}$) exists in the blockchain. Second, it validates that the certificate is still active and not expired. Finally, the smart contract verifies that the certificate is associated with the given domain. The smart contract will return a True value when everything checks out. Otherwise, it returns False.

7. The service then sends ClientKeyExchange and ChangeCipherSpec messages to negotiate the encryption algorithms and specifications that both parties will use after they established the secure channel. It then closes the handshake by transmitting a Finished message.

8. Finally, the gateway replies with a ChangeCipherSpec message to confirm the selection of encryption algorithms and specifications. It then ends the handshake by sending a Finished message.

The IoT service and gateway can construct a session key using the parameter they both got from the previously exchanged messages. Afterward, they communicate by encrypting the message with the assembled session key.

9. Once the secure channel session expires, the service votes on the gateway by executing the VoteDomain(name, score, proof) method by submitting gateway.bit as the name argument.

The score is the reputation score to give. The service can only vote once over some time, and it votes either by giving a negative or positive mark. If all verification is successful, the service gives a positive score. Otherwise, it puts a negative value. As long as most IoT services are honest, we argue that we can maintain a credible certificate reputation.

We can enforce a small payment mechanism in transaction fees or deposits to discourage malicious actors from spamming votes to a particular gateway by creating multiple fake accounts and giving

dishonest reviews. Moreover, when casting a vote, the service also discloses their `client-random` and `server-random` as `proof` of the secure channel establishment. A notary, as a trusted third-party auditor, will audit the submitted proof. The system can punish any false reporting by making the malicious voter unable to claim their deposit back.

### 3.3. Quick Analysis

Aside from single-point-of-failure and scalability issues, centralized identity management possesses other problems. First of all, CA manages trusted certificates centrally. Therefore, when those certificates are compromised, the CA cannot provide a seamless revocation procedure [34]. Moreover, the DNS has severe privacy and censorship issues. The former refers to a scenario where DNS servers can log DNS requests to get users' information on the site they are trying to request [35]. The latter happens when governments or authoritative entities censor the Internet by removing access to DNS resolvers [36].

By using our design, IoT entities register their identity to the smart contract, stored distributedly across multiple nodes in the network. Therefore, our proposal eliminates the single-point-of-failure from the system while increases overall scalability and robustness. Furthermore, because submitted public keys and domain names are replicated to each of the blockchain nodes' storage, identity lookups become local processes. This change speeds up the revocation speed since the revocation list is also cloned to all nodes. Finally, the lookups also turn private since no other party can log or censor our DNS query requests as they are now also local procedures.

## 4. Blockchain Solutions for Data at Rest

In this section, we explore integrity services for IoT data at the storage location. More specifically, we put concerns in two kinds of storage: the repository of IoT raw data and the list of IoT commands generated during IoT operations. Our main goal here is to provide a robust, non-repudiable, and tamper-proof database that can augment the value of stored IoT data and commands.

### 4.1. Motivations

We can apply solutions from our data in transit design during IoT data gathering to protect the transmitted IoT data's integrity through a secure channel. However, this protection is ephemeral as it only guards communication. In the IoT system, it is common for IoT entities to pass an IoT data or process. Once a particular entity receives data from others, the entity decrypts, stores, and continues to process it in a plaintext form. Eventually, the entity will deliver the processed data to another entity by building another secure channel. If attackers can compromise one of those IoT entities, the IoT data loses its overall credibility because our data in transit only satisfies the integrity between parties that construct the secure channel. Other entities cannot measure the quality of communication as they are not involved in it. Thus, we argue that to ensure continued integrity service, we need to provide additional protection in the IoT data itself.

In general, the digital signature algorithm, for example, the Elliptic Curve Digital Signature Algorithm (ECDSA) used in Ethereum [30], can be used to provide a non-repudiation guarantee to an IoT process. Furthermore, when multiple entities contribute to IoT operations, we can use sequential aggregate signatures [37]. Using this approach, each of the involved entities takes turn signing the process in order. In this way, they cannot deny their participation. Hossain et al. [38] proposed using a chain of signatures for forensic IoT use cases using blockchain. However, their approach requires the IoT system to store IoT data, responses, and signatures in the blockchain. Therefore, their solution is inefficient and costly.

A better idea is to only store the IoT metadata, instead of the raw data, in the blockchain. In particular, we can hash the IoT data and store only the resulted hash. Proof of existence [39] introduces this method by letting anyone anchor hashes of any digital file in the blockchain. The system puts the digital file hash in a transaction and then submits it to the blockchain. The received

transaction hash becomes the receipt for other users (acting as validators) to verify the file's authenticity. Upon receiving a digital file from the source, they can hash the file and can try to find it in the blockchain using the transaction hash. Because everyone agrees that the blockchain state remains secure, when the users obtain the digital file hash in the blockchain, they can safely assume that the file is authentic.

We propose to combine the chain of signatures and blockchain receipt techniques to provide non-repudiation and tamper-proof database for the IoT system. While designing our solution, we also conducted a literature survey on other blockchain-related solutions regarding data at rest. We then use curated papers from our survey as our design's building blocks. We summarize their contributions to our design in Table 4.

**Table 4.** The list of related works serving as building blocks for our data at rest solutions.

| Project | Related to | Contributions to our design |
| --- | --- | --- |
| FIF-IoT [38] | Chain of signatures | The author provides a role model of a chain of signatures based on blockchain. |
| Proof of Existence [39] | Blockchain Receipt | A legacy blockchain receipt proposal acts as our inspiration. |
| ChainPoint [40] | Blockchain Receipt | It provides an example of a working blockchain anchoring technique using Merkle Root hash. |
| Xueping Liang et al. [41] | Blockchain Receipt | We apply the authors' idea of using daemon to generate blockchain receipts. |
| Jay Kishigami et al. [42] | Data Sharing | The authors propose a licensing mechanism based on the blockchain that we can employ in our design to control IoT data distributions. |
| NuCypher KMS [43] | Data Sharing | We can use the proxy re-encryption technique that the authors propose to make our shared IoT data confidential. |

### 4.2. Our Proposed Solutions

**Prerequisite**: We assume that we use the distributed identity management as proposed in Section 3. All of the entities here have the secret key, public key, and address for offline and online identities. We also bring up a new entity, the auditor, which plays a role as the validator of the stored IoT data. All of the blockchain nodes in the IoT system are eligible to become an auditor. We introduce new notations that we employ throughout the rest of this paper in Table 5. Figure 4 shows an overview of our proposed data at rest design. We divide our explanations into IoT data gathering, IoT data storing, and IoT data sharing. They all contribute to the overall integrity of the stored IoT data in the database.

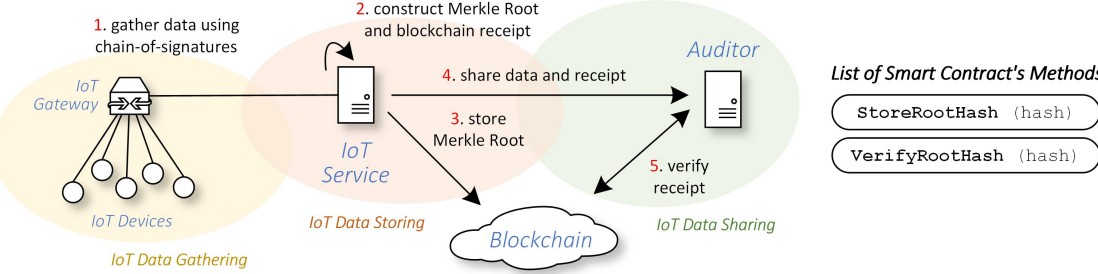

**Figure 4.** Our blockchain design to protect the integrity of stored IoT data: The protocol starts from gathering IoT data using a chain of signatures, creating Merkle Root and its corresponding blockchain receipts, and then sharing and verifying the receipts. We also depict the required list of smart contract methods for developers to implement.

**Table 5.** The list of new notations introduced in our data at rest design.

| Notation | Description |
|---|---|
| *data* | The IoT raw data from IoT devices. |
| *aggr* | The aggregated IoT data after micro-processing in the IoT gateway. |
| *comm* | The IoT commands for IoT devices from the IoT service. |
| *feed* | The feedback from IoT devices in response to IoT commands. |
| *groupID* | An identifier used to link the contents of a table in IoT gateways to the one in the IoT service. |
| $H(p)$ | A hashing operation of $p$ payload using any hash function (e.g., SHA-256 algorithm). |
| *h* | The hash of a payload, the output of $H(p)$. |
| *Y* | A table in a database. |
| *y* | A sub-table, which is a segment of a table. |
| *row* | A particular row in a table. |
| *m* | The threshold of the maximum number of new rows inserted in the table to trigger blockchain receipt generation. |
| *n* | The number of leaves as a parameter to construct the Merkle Root in the blockchain receipt. |

### 4.2.1. IoT Data Gathering

**Transmitting IoT Data**: during IoT operations, the IoT service collects IoT raw data from multiple IoT devices. The data may transit in one or multiple IoT gateways for micro-processings or as a hop to connect to the service. Therefore, multiple entities may contribute to the final IoT data. With chain of signatures, all involved parties sign the data they receive before they relay the data to other parties. Thus, when it reaches the end of the transmission pipeline, we can trace which entities have received, processed, and relayed it. In Figure 5, we outline our design of a chain of signatures usage in the IoT system, which we can further describe as follows.

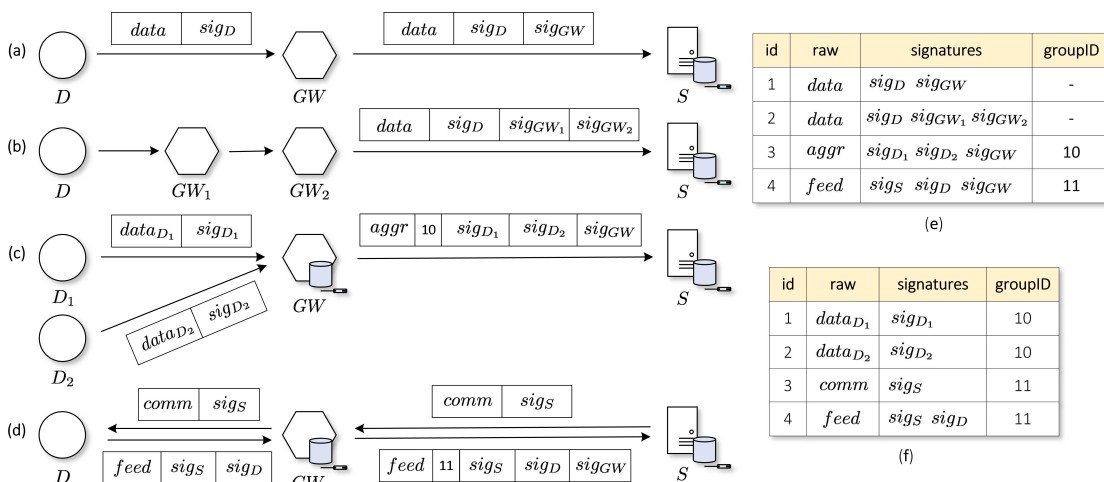

**Figure 5.** Four scenarios of IoT data flows using chain of signatures: scenarios (**a**,**b**) are the delivery of a simple IoT data reading from a single IoT device through one or multiple IoT gateways to the IoT service. Scenario (**c**) considers IoT raw data from multiple devices, including a data aggregation scheme performed by the gateway. In scenario (**d**), the IoT service transmits a command to the device, which the device acknowledges as feedback. Finally, (**e**,**f**) depict the final data stored in the service's and gateway's database from scenarios (**a**) to (**d**).

- In the first image (a), an IoT sensor device, $D$, transmits its IoT data readings to the nearby IoT gateway, $GW$. $D$ also provides its signature to the generated data, $sig_D$, which it signs using its private key, $sig_D = SIGN_{SK_D^{online}}(data)$. $data$ is an IoT data reading. $GW$ then relays the message to the IoT service, $S$. Therefore, $GW$ also appends its signature $sig_{GW}$ to the received data,

$sig_{GW} = SIGN_{SK_{GW}^{online}}(data \parallel sig_D)$. When the data arrives at the Cloud, $S$ stores signed data in its database, as shown in the image (e) with $id = 1$.

- The second image (b) depicts a similar scenario like the one in (a). However, we have two gateways now, $GW_1$ and $GW_2$. Therefore, the chains are longer compared to the previous example. When the data arrives in $S$, the service stores the received data depicted in the image (e) with $id = 2$. It contains three signatures: $sig_D$, $sig_{GW_1}$, and $sig_{GW_2}$, which were derived from $sig_D = SIGN_{SK_D^{online}}(data)$, $sig_{GW_1} = SIGN_{SK_{GW_1}^{online}}(data \parallel sig_D)$, and $sig_{GW_2} = SIGN_{SK_{GW_2}^{online}}(data \parallel sig_D \parallel sig_{GW_1})$.

- In the third scenario (c), $GW$ receives two readings from two devices: $D_1$ and $D_2$. Like the two previous scenarios, device signatures $sig_{D_1}$ and $sig_{D_2}$ accompany those readings. The signatures are obtained from $sig_{D_1} = SIGN_{SK_{D_1}^{online}}(data_{D_1})$ and $sig_{D_2} = SIGN_{SK_{D_2}^{online}}(data_{D_2})$. However, different from previous scenarios, $GW$ now has a local database. $GW$ stores $data_{D_1}$ and $data_{D_2}$ in its local database as shown in image (f) with $id = 1$ and $id = 2$. Furthermore, $GW$ also generates a reference pointer for those two data, $groupID = 10$. This pointer links contents in its local database with the one stored in $S$ later.

Once it finishes storing the data, $GW$ conducts data aggregation by averaging the values and then signs the aggregated data with its secret key, $sig_{GW} = SIGN_{SK_{GW}^{online}}(aggr \parallel groupID \parallel sig_{D_1} \parallel sig_{D_2})$. $aggr$ and $groupID$ denote the aggregated IoT data and the group identifier. After that, the gateway delivers $aggr$, $sig_{GW}$, $sig_{D_1}$, $sig_{D_2}$, and $groupID$ to $S$. The service then saves them in the database, as shown in the image (e) with $id = 3$.

**Transmitting IoT Command**: In our final scenario, we use a chain of signatures to ensure the integrity of IoT commands issued from IoT services to IoT devices, depicted in Figure 5 in the image (d). $S$ creates an IoT command, *comm*, for the device. The service signs this command with its secret key, $sig_S = SIGN_{SK_S^{online}}(comm)$. It then delivers the command and the associated signature to the specified IoT gateway, where the device resides.

Upon receiving this command, $GW$ verifies that $sig_S$ is valid. The gateway then creates another reference pointer for this process, $groupID = 11$, and stores the command in local storage, shown in image (f) with $id = 3$. After that, $GW$ relays the command to the IoT device. We assume that the device fully trusts the gateway and that all communications from the device will return to the service through the same gateway. Therefore, the gateway does not need to include the signature at this moment. After processing this command, the device sends acknowledgment as feedback. The device forms a feedback, *feed*, and its signature, $sig_D = SIGN_{SK_D^{online}}(feed \parallel sig_S)$. It then sends them back to $GW$ by also including the IoT service's original signature, $sig_S$.

$GW$ stores the feedback and signatures in its database, shown in image (f) with $id = 4$. After that, the gateway signs this feedback, $sig_{GW} = SIGN_{SK_{GW}^{online}}(feed \parallel sig_S \parallel sig_D)$. $GW$ then deliver *feed*, $groupID$, $sig_S$, $sig_D$, and $sig_{GW}$ to $S$. Finally, the feedback and signatures rest in $S$'s database as in image (e) $id = 4$.

Using a chain of signatures, we can pinpoint the origin of the IoT data or command stored in a database, which is the first instance that signs that data or command. It also provides strong non-repudiation properties for all involved participants. However, signing and verifying digital signatures are costly operations in terms of CPU resources. Therefore, administrations may limit these procedures only to essential or crucial IoT operations.

### 4.2.2. IoT Data Storing

The chain of signatures only provides a partial integrity guarantee to our databases. Assuming that attackers can compromise the service or gateway, they can further tamper the storage and render those signatures obsolete. Therefore, we need a complementary tamper-proof property in our database. Using blockchain is one solution, but it is expensive to store all the data,

commands, pointers, and signatures in the blockchain. Proof of existence [39] introduces the idea of storing the hash of data instead of the raw data in the blockchain. As a result, we can reduce the byte size that has to be saved. However, this method is still inefficient because the IoT system generates a vast amount of data. To save much more storage space, we can extend the hashing further by forming a Merkle Tree [44] and store only the root hash in the blockchain [40].

**Generation of Blockchain Receipt**: The blockchain returns transaction (tx) hashes each time we store data in it. These tx hashes are receipts or proof that a particular data exists in the blockchain, thereby maintaining the data's credibility because the blockchain is tamper-proof. We can leverage these so-called "blockchain receipts" to build a verifiable database system. Figure 6 showcases our overall blockchain receipt design for an IoT system. We describe the generation of receipts in the following paragraph and elaborate on its verification in the next section.

**Figure 6.** (**a**) A scenario for constructing a Merkle Root, $h_7$, from a table containing 400 rows of IoT raw data ($m = 400$) split into 4 sub-tables ($n = 4$): From this structure, we can build blockchain receipts for each of the sub-tables. (**b**) The receipt for $y_2$.

1.  First of all, we have two essential parameters for blockchain receipts: $m$ and $n$. $m$ is a threshold defining the number of newly inserted rows in a table to trigger the blockchain receipt generation process. Meanwhile, $n$ is the number of leaves to construct the Merkle Root [44]. These two parameters are configurable by the administrations. For simplicity, let us assume that we set $m$ equal to 400 and $n$ to the value of 4.

2.  During IoT operations, the IoT service receives and saves IoT data in its database. Suppose that we have a daemon in our system that detects our storage state, similar to the idea proposed in [41]. Once the service has more than 400 newly inserted data, it triggers the $m$ parameter and starts building blockchain receipts.

3.  Following the $m$ parameter, the service queries 400 rows of tables from the database. We separate these data into a new table, $Y$. Then, we divide this table into 4 sub-tables, $Y = \{y_1, y_2, y_3, y_4\}$, matching the $n$ parameter. $y_1$ is a concatenation of the first 100 rows in the table, which is $y_1 = row_1 \parallel row_2 \parallel ... \parallel row_{100}$. Meanwhile, $y_2$, $y_3$, and $y_4$ are for the second, the third, and the fourth 100 rows. They are $y_2 = row_{101} \parallel row_{102} \parallel ... \parallel row_{200}$, $y_3 = row_{201} \parallel row_{202} \parallel ... \parallel row_{300}$, and $y_4 = row_{301} \parallel row_{302} \parallel ... \parallel row_{400}$, respectively.

Figure 6a illustrates this table division. For easy explanation, the picture only shows a small table data structure with two columns: id and data. However, in real environments, we can have a higher number of rows and columns. More importantly, the tables must also include the column that stores our chain of signatures from the previous section, which can augment the data's authenticity in each row.

4.  The service calculates the Merkle Root hash, which is again illustrated in Figure 6a. From each of the sub-tables, the service generates the hash: $h_1 = H(y_1)$, $h_2 = H(y_2)$, $h_3 = H(y_3)$, and $h_4 = H(y_4)$. $H(.)$ refers to any hash function. After that, the service hashes $h_1$ and $h_2$ to generates $h_5$. Then, it hashes $h_3$ and $h_4$ to produces $h_6$, specifically $h_5 = H(h_1 \parallel h_2)$ and $h_6 = H(h_3 \parallel h_4)$.

Following a pyramid scheme, the service then forms the summit by hashing $h_5$ and $h_6$ together: $h_7 = H(h_5 \parallel h_6)$. This $h_7$ is the Merkle Root hash.

5.  The service then stores $h_7$ in the blockchain by forming a transaction to call the `StoreRootHash` (`hash`) method in the smart contract. The `hash` is the Merkle Root hash to be stored, which is $h_7$.

6.  Once the service saves the root hash in the blockchain, it generates blockchain receipts for each of the sub-tables. An example of such a receipt is shown in Figure 6b. That receipt is for $y_2$, which contains data from row 101 to 200. In the receipt, we can find the *m*, *n*, *root*, and *proof* information. The *root* indicates the Merkle Root hash, while the *proof* is the trace of the $y_2$'s opposite hash in the pyramid scheme to reach the root hash. Then, the service keeps these blockchain receipts in a secure place separated from IoT raw data storage.

### 4.2.3. IoT Data Sharing

**Verification of Blockchain Receipt**: in most cases, the IoT service needs to allocate its stored IoT data and command to other entities for analytics or presentations. The following paragraphs explain how the second entity, as an auditor, can use the blockchain receipts to validate an arbitrarily assigned data and can determine whether anyone tampered with the data.

1.  The IoT service delivers the IoT data, *Y*, and its corresponding blockchain receipts to the auditor. In this example, we assume that the service shares the data in plaintext form and without any authentication.

Depending on its contents, we might need to protect data privacy and only allow limited access to the data. In this case, we can borrow the licensing mechanism proposed in [42]. Using their approach, we can set licenses or permissions to each of the allocated IoT data only to registered, trusted, or paid customers.

Moreover, to further enhance data privacy, we can also conduct encryption procedures to the IoT data. We are interested in using the proxy re-encryption technique proposed in [43]. This novel encryption technique allows an entity to transform (by re-encrypting) an encrypted data from one public key to another, without decrypting it. Combined with the licensing technique, we can then provide fine-grain access control to each of the IoT data, while our chain of signatures and blockchain receipt proposal preserves the integrity of the data.

2.  After receiving all of the data and associated receipts, the auditor queries the *m* and *n* parameters from the receipt. First, it takes 400 records of data following the *m* parameter. Then, because *n* is equal to 4, the auditor splits the data into 4 sub-tables and forms $y_1'$, $y_2'$, $y_3'$, and $y_4'$. For simplicity, we only show the verification of $y_2'$ in this section. The corresponding blockchain receipt is depicted in Figure 6b.

3.  $y_2'$ contains the second 100 rows of the table: $y_2' = row_{101}' \parallel row_{102}' \parallel ... \parallel row_{200}'$. The auditor hashes $y_2'$ to generate $h_2'$, $h_2' = H(y_2')$.

4.  Afterward, the auditor queries the *proof* information from the receipt. The *proof* contains instructions on aligning the provided corresponding hashes ($h_1$ and $h_6$) with $h_2'$ to form the Merkle Root hash. The auditor must follow the directions in order. The first line of the *proof* gives two parameters: the hash $h_1$ and the hash's alignment *left*. Therefore, the auditor puts $h_1$ on the *left* of $h_2'$ and hashes it to generate $h_5'$, $h_5' = H(h_1 \parallel h_2')$. After that, following the second line guidance, the auditor puts $h_6$ on the *right* of $h_5'$ to generate $h_7'$: $h_7' = H(h_5' \parallel h_6)$. When we find no other advice, it means that we reached the Merkle Root hash already. In this case, $h_7'$ will be the auditor's generated root hash.

5.  The auditor then queries the *root* information from the receipt and obtains $h_7$, the Merkle Root generated by the IoT service.

6.  The auditor then conducts two verifications. First, it needs to ensure that $h_7'$ is equal to $h_7$. Afterward, it needs to check if this $h_7'$ is recorded in the blockchain. The auditor forms a transaction that calls the `VerifyRootHash(hash)` method in the smart contract. The `hash` is the Merkle Root

hash to be verified, which is $h'_7$. The method will return `True` if they find $h'_7$ in the smart contract's storage.

7. If all of the above validations do not result in an error (or return `False`), then the auditor can assure that no one tampered with the sub-table data's content. The auditor can then continue the same process to verify the rest of the sub-tables, mainly $y'_1$, $y'_3$, and $y'_4$.

### 4.3. Quick Analysis

Luo et al. [45] elaborate a data poisoning scenario for IoT models. In their scenario, attackers have access to several compromised IoT devices. Then, they create fake or false inputs for the systems to confuse or misclassify the IoT systems. Without a non-repudiable and tamper-proof storage, the IoT administration may find difficulties detecting the origin of the attacks. They may guess that the breach happens at the IoT service where the IoT data is stored. They can also randomly speculate that the IoT gateway is acting maliciously by sending invalid data. Our proposal augments IoT databases to have non-repudiable and tamper-proof properties. Therefore, we can speed up this detection. The administration can swiftly verify whether the system's databases are secure and quickly determine that the IoT devices themselves are hostile.

During the collaborations between two parties in our system, one entity can become malicious, either intentionally or unintentionally. Let us say that the first party cheats the agreed SLA by sharing fake or invalid data with the second party. Therefore, their action is damaging the receiving party. The second party can sue the first party by bringing the log of IoT data and commands as evidence to the court. However, without a non-repudiable and tamper-proof database, the submitted evidence will lose the "forensically sound" property [46] and no longer have credibility. In contrast, with the combination of the chain of signatures and blockchain receipts that we propose, we can preserve the log's integrity to guarantee its trustworthiness.

## 5. Blockchain Solutions for Data in Process

We look at two integrity services during IoT data processing. In the collaborative IoT data flow pipeline, we are interested in the alliance between IoT services and workers by forming a decentralized marketplace. Meanwhile, we consider designing a federated learning platform in the blockchain for the privacy-preserving IoT data flow pipeline.

### 5.1. Motivations

**Collaborative IoT data flow pipeline**: The vast amount of IoT data that the IoT services gathered are idle in their database, waiting for IoT workers to process them to generate insights. It is simple to let a particular IoT application deploy its own workers to process their data. However, this solution is expensive and unproductive. In particular, when the application has no more data to process, then its workers will contribute nothing to the system. Therefore, nowadays, shared IoT data processing services such as the one in Amazon [47] and Azure [48] are gaining popularity. Instead of building their own silos and training the IoT raw data in their private servers, IoT services now can rent other companies' servers and instruct them to train the IoT services' data.

To facilitate the previously mentioned collaboration scheme, we need an electronic marketplace (e-marketplace). We can leverage blockchain as a platform to build a reliable marketplace. For example, Sterling [49] showcases a decentralized marketplace based on blockchain. The authors combine blockchain smart contracts, trusted execution environment, and differential privacy to provide a secure and privacy-preserving marketplace. Our data in process design goal is similar to this proposal. However, unlike Sterling, we do not consider the IoT raw data and machine learning models as commodities in our market. Instead, we use the market as tools for IoT workers to exchange IoT training services among themselves.

**Privacy-preserving IoT data flow pipeline**: the practice of companies building private data silos and then harvesting as much data as possible from users is deteriorating users' privacy. Users are not

in control of their data, and most of the time, they do not have any options but to comply with this malpractice simply because they want to reap the "free" service that the company offers. We argue that this problem is by design; we cannot expect the IoT device to generate analytics by itself. The device needs processing power, and much data from other IoT devices, which with the current design, is very impractical. Nowadays, Federated Learning [50] (FL) is gaining traction because of the ability to train data in the IoT device's hardware with limited data by collaborating with other IoT devices. Instead of sharing private data, the device shares the trained model to other entities, which they can aggregate to construct the global trained model. However, the original FL architecture is still centralized because the IoT service solely manages the training. Therefore, we can leverage blockchain as a decentralized platform to conduct FL.

CrowdSFL [51] proposes a crowdsourcing platform that leverages blockchain as a marketplace for requesters (analogous to IoT services) and workers. However, in their approach, the requester solely updates the global model. Thus, it holds a decisive role in determining the shape of the trained model. As a result, this proposal is strongly centralized. Yang Zhao et al. [52] also suggested a similar blockchain platform to perform FL, which makes use of Mobile Edge Computing (MEC) servers as helpers for IoT devices to train their data. They propose using the Algorand-based leader election scheme [53] to choose a candidate miner to update the global model. This approach is quasi-centralized because the algorithm picks only small subsets of participants to become the leader. Unlike those proposals, we design our FL protocol to involve all IoT devices to simultaneously maintain their own view of the global model. Thus, our design is decentralized.

While devising our solutions, we also perform a literature survey on other blockchain researches related to e-marketplace and federated learning. We then employ curated papers from our survey as our design's building blocks. We summarize their contributions to our design in Table 6.

**Table 6.** The list of related works serving as building blocks for our data in process solutions.

| Project | Related to | Contributions to Our Design |
|---|---|---|
| Sterling [49] | E-marketplace | This project shows us an example of a privacy-preserving blockchain-based marketplace for private data and machine learning model trading. |
| Wibson [54] | E-marketplace | We can use the notary system that the authors propose to be auditors and mediators between data sellers and data buyers in our marketplace. |
| CrowdSFL [51] | Federated Learning | We can employ the reward distribution mechanism and the El-Gamal re-encryption scheme in this proposal to our design. |
| Yang Zhao et al. [52] | Federated Learning | We borrow the idea of using MEC servers to alleviate some training burdens from IoT devices. |
| Dongxiao Liu et al. [55] | Reputation System | We endorse the use of the proposed reputation system, which provides an anonymous reputation system that is hard to trace to protect the reviewers' privacy. |
| Jiawen Kang et al. [56] | Incentive Mechanism | We use the proposed incentive mechanism, which presents an incentive mechanism for mobile devices to encourage active participation in the federated learning process. |

*5.2. Our Proposed Solutions*

We divide our explanations into two themes: the decentralized marketplace and federated learning. Both of them are based on blockchain and contribute to the IoT data processing integrity. We bring up new notations regarding our data in process design in Table 7.

<div align="center">

**Table 7.** The list of new notations introduced in our data in process design.

</div>

| Notation | Description |
|----------|-------------|
| $MEC$ | The Mobile Edge Computing (MEC) servers. |
| $h_S$ | The Merkle Root hash of the IoT service dataset, which also acts as job identifier in our marketplace. |
| $prop$ | A training proposal from IoT worker to IoT service. |
| $r_W$ | The training result from an IoT worker. |
| $global$ | The global model from an IoT service for federated learning. |
| $h_{IPFS}$ | The IPFS hash of our global model, which serves as training process identifier in our federated learning. |
| $local$ | The updated local model from an IoT device. |
| $h_{D_a}^b$ | The IPFS hash of a local model from device $a$ for epoch $b$. |

### 5.2.1. Blockchain-Based Decentralized Marketplace

**Prerequisite**: first of all, we use our distributed identity management, as proposed in Section 3. Entities have the secret key, public key, and address for both offline and online identities. Then, we introduce several actors.

1. IoT services act as buyers in our e-marketplace. We assume that they have many raw IoT data from their IoT operations, but they cannot train them on their own.
2. IoT workers exist as sellers in the market, which provides training services for IoT services.
3. A notary (e.g., the government as a trusted third-party) initially governs the market. This entity develops the market rule by also counting feedback from IoT services and workers as the market players. The market can be run automatically in a decentralized manner without any further intervention once the notary deploys the smart contract.

Afterward, we made the following assumptions.

1. The smart contract has a reputation system, and the IoT services and workers already built some positive scores in the system.

   In general, we can integrate our market with any available reputation system. However, we are interested in using the proposal from [55]. The authors present a reputation procedure with the anonymity feature, which can protect users' privacy. Therefore, users are most likely to give honest reviews without being worried about being tracked or discriminated against if they give poor score reviews.
2. In this scenario, we use deep neural networks [57] as an example of IoT data training. However, this market can be redefined customarily to match any machine learning algorithm. Thus, it is algorithm-agnostic.
3. For simplicity, we only show a one-to-one mapping scenario between IoT services and workers. In real cases, race conditions may exist. IoT services can contact many IoT workers, and IoT workers may process multiple jobs from several IoT services.

Figure 7 summarizes a scenario where the IoT service negotiates an IoT training service with the IoT worker. We elaborate on them in the following paragraphs.

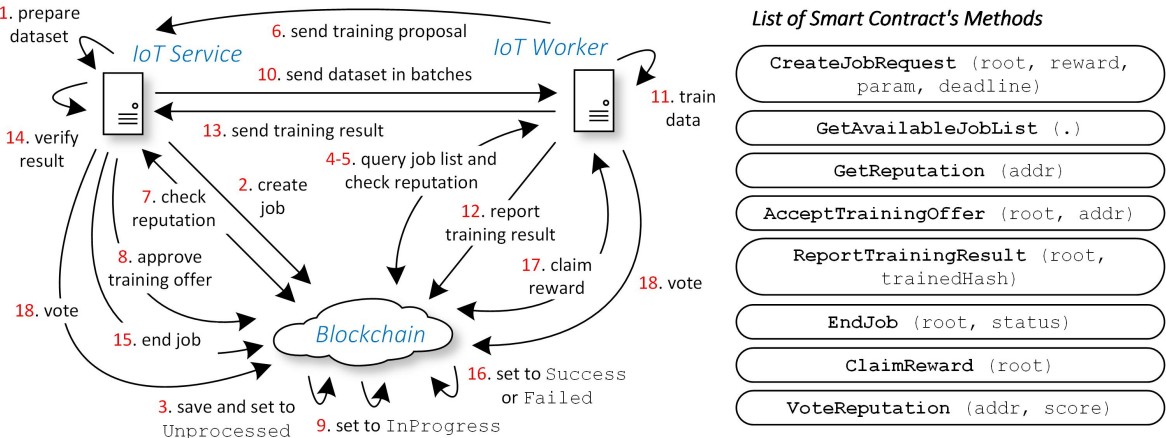

**Figure 7.** Overview of our blockchain design of the decentralized marketplace: the design flow starts from an IoT service creating a training job offer for IoT workers. The workers then propose a training offer. If negotiation is complete, the worker trains IoT service's data and submits reports back to the service and blockchain. Finally, the worker claims their reward when the job finishes. The figure also depicts a list of smart contract methods for developers to implement.

1. The IoT service conducts preparation for IoT data processing. It transforms its whole IoT raw data into a training dataset and splits them into several data chunks (or batches). During this separation, the IoT service applies our data at rest strategy in Section 4 to generate Merkle Root hash and the blockchain receipts. More specifically, $Y$ becomes the whole training dataset, and $y$ is the training batch. Let us also say that $h_S$ is the root hash of the training dataset.

2. The IoT service then creates a job request in the smart contract by calling the `CreateJobRequest(root, reward, param, deadline)` method. `root` is the Merkle Root hash of the training dataset, $h_S$. `reward` is the payment for IoT workers after completing the task, e.g., 100 coins. `param` is the training parameter and requirements. They can be the type of training (e.g., classification, prediction, or generation) and the desired training result (e.g., having at least 90% accuracy). Finally, `deadline` is the time limit for the workers to return the training result to the service, e.g., a UNIX timestamp.

   In this step, the IoT service also pays the reward in the form of a deposit. This policy ensures that the service indeed has the money to pay the workers; therefore, they can trust the job offer.

3. The smart contract maintains lists of job requests in local storage. At any given time, they can only have one of the following states in the smart contract.

   - `Unprocessed` is when the IoT service just offered the job request, and no IoT worker processes it.
   - `InProgress` tells us that the IoT service has assigned the job to one of the IoT workers.
   - `Success` indicates that the job request is completed and that both the IoT service and worker are satisfied with the result.
   - `Failed` implies that the job's processing is unsuccessful due to one of the party cheats or timeout.

   Upon receiving a transaction from the previous step, the smart contract queries the sender information from the transaction, which is $\alpha_S^{online}$, then combines it with arguments from the method and saves them to the list. At this moment, the smart contract labels this new request as `Unprocessed`.

4. IoT worker queries for available job lists in the smart contract by invoking the `GetAvailableJobList(.)` method. The smart contract returns a list of all jobs that currently have the status of `Unprocessed`. Each of the available jobs is distinguishable by `root` or $h_S$ as the job identifier.

5. Optionally, the worker can check the IoT service reputation by calling the `GetReputation(addr)` method in the smart contract. `addr` is the address of the IoT service, $\alpha_S^{online}$, which the worker obtains when querying for the available job list in the previous step. Based on this reputation, the worker can decide whether to trust the job offer.

6. The worker picks one of the jobs that it is willing to train. In general, it can pick any job with the highest reward to maximize profit. However, the worker should choose wisely with consideration of its processing ability. If the worker fails to report the desired output to the service before the deadline, the worker will not be able to claim the reward, and this event can diminish its reputation.

After deciding which job to take, the worker delivers a training proposal to the IoT service off-chain. We denote *prop* as the training proposal. The worker first includes the job identifier, $h_S$, and its address, $\alpha_W^{online}$, in the *prop*. Then, it also must provide their signature, $sig_W = SIGN_{SK_W^{online}}(prop)$. Therefore, the final proposal will be $prop \leftarrow h_S \parallel \alpha_W^{online} \parallel sig_W \parallel prop$.

7. Upon receiving the proposal, the IoT service validates the worker's signature, and $VERIFY_{PK_W^{online}}(prop, sig_W)$ must return `True`. After that, the IoT service can check the worker's reputation by calling the `GetReputation(addr)` method. `addr` is the worker's address, $\alpha_W^{online}$. The acceptance of a job request is then subject to the service's judgments over the worker.

8. Assuming that the service is satisfied with the worker's reputation, the service accepts the training offer by calling the `AcceptTrainingOffer(root, addr)` method. `root` and `addr` are the job id, $h_S$, and the worker's address, $\alpha_W^{online}$, respectively.

9. The smart contract sets the job request to `InProgress`. While in this state, no other workers except the one that the IoT service previously approved can handle this job. This lock-in mechanism ensures that no race conditions happen to boost the fairness of the market.

10. The service sends batches of raw IoT data to the worker off-chain. On each batch, the service also includes the associated blockchain receipt. With this receipt, the worker can verify that the incoming batches are part of $h_S$, as described in Section 4. In other words, the service cannot cheat the worker by training more data than they previously registered in step 2.

11. The worker begins training the received data.

12. After it is complete, the worker hashes the training result (e.g., the checkpoint data if we are using TensorFlow [58]) and stores it in the smart contract. We denote $r_W$ as the training results from the IoT worker and $h_W$ as its hash, $h_W = H(r_W)$. The worker then calls the `ReportTrainingResult(root, trainedHash)` method, with `root` and `trainedHash` as the job identifier, $h_S$, and the hash of training result, $h_W$.

13. The worker delivers the training result to the service off-chain. Before transmission, the worker must include the job identifier, $h_S$, and its signature, $sig_W = SIGN_{SK_W^{online}}(r_W)$ to the training result. The final training result will be $r_W \leftarrow h_S \parallel sig_W \parallel r_W$.

14. The service verifies the signature and checks if the hash of the received training result matches the one in the smart contract. Specifically, $VERIFY_{PK_W^{online}}(r_W, sig_W)$ must return `True` and the service can find $h_W$, calculated from $h_W = H(r_W)$ in the smart contract. Furthermore, it makes sure that the result fulfills the desired training output agreed during the job request submission in step 2. Note that the service can replicate (without retraining) the training process easily with the given checkpoint to determine whether the training indeed has the desired output.

15. Assuming that the service is satisfied with the result, it ends this job request by calling the `EndJob(root, status)` method. The `status` is either `Success` or `Failed`. If the result fulfills the training objectives, then the service sends `Success` state. Otherwise, they set the status to `Failed`. As a fail-safe mechanism, the service can only call this method after the worker submits the training result. Therefore, the service cannot cancel `InProgress` jobs arbitrarily and damage the worker's training efforts.

16. Upon receiving the previous transaction, the smart contract sets the job request to either `Success` or `Failed`.

17. When a job request is successful, the worker withdraws the `reward` from the smart contract using the `ClaimReward(root)` method. However, if the job fails, the service takes its previous deposit back from the smart contract using the same method. Moreover, the service can also retrieve their deposit back if the worker cannot complete the training within the given `deadline`. The smart contract will automatically set the job status to a `Failed` state.

18. Once the job session ends, whether it is successful or failed, the service and worker vote for each other's reputation by invoking the `VoteReputation(addr, score)` method. `addr` is the address of the service or worker, $\alpha_S^{online}$ or $\alpha_W^{online}$. Meanwhile, the `score` is the reputation value to give.

**Notary services**: because the market is decentralized, it opens the possibility for each player to cheat the system. The worker can intentionally send a fake result in step 13 to halt IoT servers to get a valid training result. Recall that the system locks the training process of an `InProgress` job request only to one worker at any given time. As a result, if the worker is malicious, it can delay the service time to get a valid training result. Moreover, the service can also deliver a fake confirmation in step 15. Even though the worker already transmitted a valid training result, the service can maliciously deny this result by sending fake approval. To solve these issues, we need the help of a trusted third party. For example, we can employ a notary service [54] as a mediator and judge in case of conflicts. This notary is the same entity that deploys the smart contract.

### 5.2.2. Blockchain-Based Federated Learning

**Prerequisite**: similar to our decentralized marketplace design, here we also apply our distributed identity management from Section 3. All entities have the secret key, public key, and address for both offline and online identities. The following actors are present in our federated learning.

1. IoT devices generate IoT raw data and train them in their local machine. Instead of allocating their private IoT data, IoT devices share their training results with IoT services.

2. Mobile Edge Computing (MEC) servers are edge servers available near IoT devices. They help to alleviate parts of the training from the IoT devices by splitting the training model.

3. IoT services take the trained model from IoT devices and provide analytics to IoT users. They also create the training rule and apply it to the smart contract. If IoT devices agree with the regulation, they can join the training.

We made the following assumptions.

1. The smart contract has a reputation system, and IoT devices and the service have positive scores. We use the same reputation procedure [55] as in our decentralized marketplace.

2. In this example, we apply the federated learning that McMahan et al. [50] proposed. However, our design can be reconstructed to match any federated learning algorithm. Thus, it is algorithm-agnostic.

3. We only show a one-to-many mapping scenario between the IoT service and their manufactured devices. In real cases, several IoT services can use the platform simultaneously.

The following steps detail our federated learning design, which we also summarize in Figure 8.

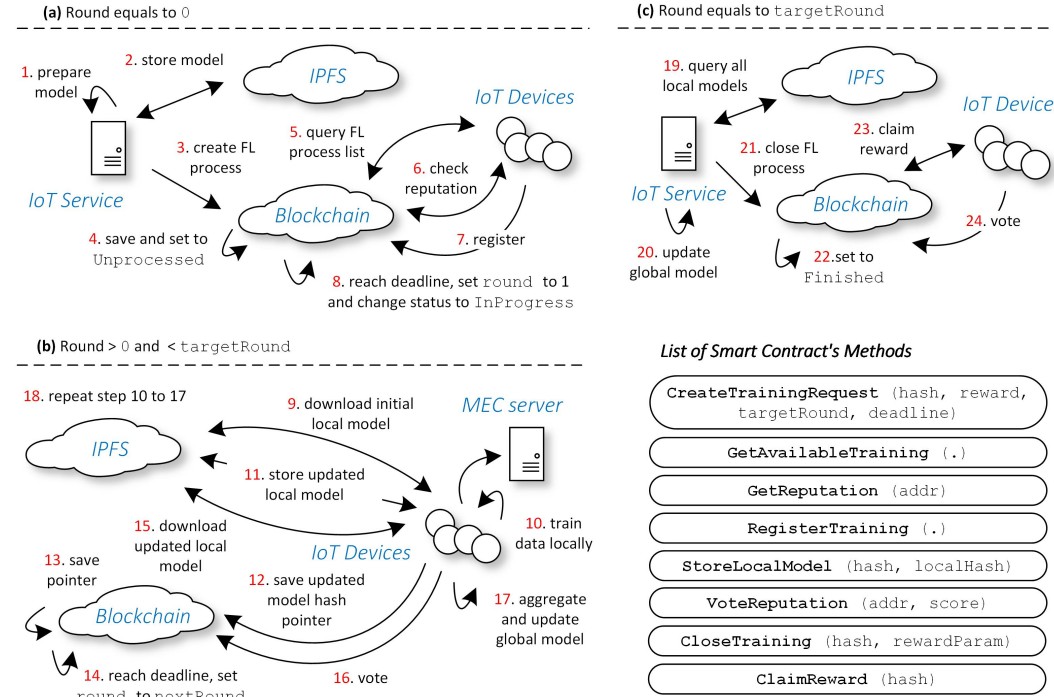

**Figure 8.** The overview of our blockchain design for federated learning: IoT devices first download the initial model from the IoT service. At each epoch, they train the model using private IoT data, share the result to other devices, and update the global model. Once the training is complete, the IoT service delivers rewards to IoT devices based on their contributions. We also show the list of required smart contract methods for developers to implement.

1. The IoT service prepares the model and its parameters (e.g., the number of nodes and layers, the structure of the model, the value of learning rate, weight, bias, and the activation function to use).

We define two models: local and global. The former is a model that the IoT devices train independently without considering private data from other devices. Meanwhile, the latter is an aggregation of local models, where IoT devices combine all local model parameters from others.

2. The IoT service signs the model and obtains the signature, $sig_{global}$, derived from $sig_{global} = SIGN_{SK_S^{online}}(global)$. The *global* denotes the initiated global model. The IoT service then stores this signature and the global model in the InterPlanetary File System (IPFS) [59]. From this action, the service retrieves the IPFS hash, $h_{IPFS}$, indicating that the IPFS network has saved its model.

3. The IoT service then creates a new federated learning process in the blockchain by calling the `CreateTrainingRequest(hash, reward, targetRound, deadline)` method. `hash` is the IPFS hash of the model, $h_{IPFS}$, which also acts as a training process identifier to distinguish one training from another. `reward` informs the number of payments for all voluntary training participants (e.g., 10 coins). `targetRound` denotes the final global epoch required to end this federated learning process (e.g., 3 epochs). Finally, `deadline` is the time limit range (e.g., in UNIX timestamp) for IoT devices to submit their local model at each epoch.

In this step, the IoT service must pay the reward in the form of a deposit. This deposit testifies that the payment is indeed redeemable, and it can enforce trust among training participants.

4. The smart contract maintains lists of training requests in local storage. At any given time, they can only have one of the following states in the smart contract.

- `Unprocessed` tells us that the training request is open for anyone to join.
- `InProgress` informs us that the training registration has finished and that all participants are currently performing training.
- `Finished` indicates that the IoT service has ended this request.

After receiving the transaction from the previous step, the smart contract saves the transaction parameters in its key-value storage. It also stores the sender of the transaction, $\alpha_S^{online}$. The smart contract then sets the `round` parameter to 0. This `round` tells the current training epoch the federated learning algorithm. By assigning this parameter, the smart contract also initiates this training request's status as `Unprocessed`.

5. Interested IoT devices can begin querying for available federated learning processes using the `GetAvailableTraining(.)` method. The smart contract returns a list of all `Unprocessed` training requests, which includes the owner address, $\alpha_S^{online}$ and the corresponding training identifier, $h_{IPFS}$.
6. Optionally, IoT devices can check for the training request owner's reputation by calling the `GetReputation(addr)` method. `addr` is the address of the IoT service, $\alpha_S^{online}$. If IoT devices satisfy the service's reputation, they can continue the process. Otherwise, they can stop and look for other training requests.
7. After choosing one training request to perform, IoT devices register to participate in the training by invoking the `RegisterTraining(.)` method. The smart contract will take the sender of this transaction, $\alpha_D^{online}$, and will save it to the list of participants in its local storage.
8. The `deadline` for the current epoch expires. The smart contract then sets the status of the training request to `InProgress` and the `round` parameter to 1. With this change, registration is now closed. The devices can now begin performing training.
9. IoT devices download the initial global model from the IPFS network using $h_{IPFS}$. They also need to verify the authenticity of the model, and $VERIFY_{PK_S^{online}}(global, sig_{global})$ must return `True`.
10. They then perform the local training using their own machines and private data. Once the training is complete, they update their local model parameters.

Because IoT devices are constrained devices, they may outsource some training to the MEC servers. Yang Zhao et al. [52] propose to train the fully connected layers in those servers while feature extractions are done in IoT devices. They also apply the differential privacy parameter between those layers to prevent the leak of private data to MEC servers.

When using this feature, IoT devices must ensure that they communicate with valid MEC servers that the IoT service endorses. Therefore, additional handshakes using our proposal in Section 3 are required. During registration, the IoT service can sign its MEC server address, $sig_{MEC} = SIGN_{SK_S^{online}}(\alpha_{MEC}^{online})$, and then present this signature as a certificate. In this case, the IoT service behaves like a CA for MEC servers.

11. IoT devices sign their updated local model and generate the signature, $sig_{local} = SIGN_{SK_D^{online}}(local)$. $local$ denotes the updated local model. They then upload the model and its signature in the IPFS network. We denote $h_{D_q}^b$ as the IPFS hash containing the local model update from device $a$ for epoch $b$. Thus, $h_{D_1}^1$ and $h_{D_2}^1$ represent the first and second IoT devices' hash for the first epoch.

For simplicity, the updated model is transmitted in plaintext form. However, encryptions can be enforced when necessary.

12. IoT devices then store the local update metadata to the blockchain by calling the `StoreLocalModel(hash, localHash)` method. `hash` is the training identifier, $h_{IPFS}$. `localHash` is the IPFS hash pointer for others to download the updated local model (e.g., $h_{D_1}^1$).

13. When receiving the transaction for this method, the smart contract put the hash and sender of this transaction (i.e., $\alpha_D^{online}$) into storage. The smart contract maintains a list of the updated local model per epoch.
14. The `deadline` for the current epoch expires. The smart contract sets the `round` parameter to 2.
15. IoT devices are aware of the round change and begin downloading all updated local models from other devices. They query for all models from $h_{D_1}^1$ to $h_{D_n}^1$, where $n$ denotes the index of the last IoT device.
16. Each of them validates the downloaded model and votes for the quality of the model. They make sure that $VERIFY_{PK_D^{online}}(local, sig_{local})$ returns `True`. After that, they call `VoteReputation(addr, score)` with `addr` and `score` as the device's address and the reputation score to give.

This voting is essential to determine and punish malicious devices that perform poorly. If a particular device's score is inferior, the system can exclude its model from being combined with others. Thus, we can preserve the grade of the global model. Furthermore, IoT services will only allocate rewards to devices that operate well during training.

17. IoT devices simultaneously aggregate all local model parameters from all devices and update the global model.
18. IoT devices then perform training for the next epoch (repeating step 10 to 17) until `round` matches the `targetRound` that the IoT service specified previously in step 3.
19. Once the final epoch is complete, the IoT service gathers all of the local model parameters from devices.
20. After that, it updates the global model.
21. The IoT service starts to distribute the reward to all IoT participants by invoking the `CloseTraining(hash, rewardParam)` method. `hash` is the training identifier, $h_{IPFS}$. `rewardParam` is a softmax array multiplier that will be applied to split compensation to all training participants.

We assume that the IoT service has a strict and fair policy on allocating rewards to IoT devices. In particular, it may use the proposal in CrowdSFL [51], where the price is given based on the number of private data used in training and the quality of the training. Otherwise, other incentive mechanisms such as [56] can be used.

22. The smart contract then changes the training status to `Finished`.
23. IoT devices can begin claiming their reward using the `ClaimReward(hash)` method with the `hash` sets to $h_{IPFS}$.
24. Last but not least, the IoT device can vote for the IoT service using the `VoteReputation(addr, score)` method. `addr` and `score` are the IoT service address, $\alpha_S^{online}$, and the reputation score to give.

### 5.3. Quick Analysis

The state-of-the-art e-marketplace is centralized and poses many obstacles. The classic problems commonly found in a centralized system are single-point-of-failure and scalability issues (especially during sales period [60]). Furthermore, we also notice monopoly issues. Because of the central control of the marketplace, the administrator can change the policy that may benefit themselves but damage users in the market. For instance, Uber changes their pricing policy several times that may put either drivers or riders at a disadvantage [61]. This issue is augmented by the fact that the market most probably is not transparent. The publics' remarks or suggestions are most likely not included in the market policy's decision-making. Using our designed marketplace, the market's rule is deterministic and transparent because the notary implements it as a smart contract. Moreover, the blockchain is fully distributed; therefore, it can further scale the market.

The prevailing FL faces similar centralization problems as in the e-marketplace. Even though the device sends only the model and not the raw IoT data to the server, the server still plays a crucial

role in determining the training result when forming the global trained model. Thus, we can expect another monopoly scheme if the server for a particular reason does not include or accept the submitted trained model from a particular IoT device [62]. Using our designed FL, IoT devices can construct the global model independently without IoT service intervention. We can then preserve the generated global model's fairness because the FL rule is reflected in the smart contract code, which is open to the public and immutable. Last but not least, the blockchain's tamper-proof properties are useful to serve as a piece of strong evidence to solve any dispute among entities that may happen in our market and FL process.

## 6. Discussion

In this section, we present how our proposed blockchain integrity platform model can solve the previously mentioned open problems in IoT big data management. We then elaborate on several further considerations and challenges in realizing our proposed design.

### 6.1. Solutions to IoT Big Data Open Problems

A previous survey study has investigated several open problems regarding the IoT big data architecture [63]. We selectively pick issues related to data in transit, data at rest, and data in process. Then, we discuss our contributions in solving the mentioned obstacles, as shown in Table 8. Overall, our design answers several of those open problems, especially for security-related ones. Our proposal also assists the non-security issues by augmenting the integrity of their solutions.

**Table 8.** Summary of how our proposed design can tackle open problems mentioned in [63].

| No | Open Problems | Related to | Our Solutions |
|----|---------------|-----------|---------------|
| 1 | Users are reluctant to rely on conventional IoT big data systems because they do not provide reliable SLA. | Data in Process | We transform SLAs into smart contracts, and all users can safely assume that the execution of a smart contract is always deterministic; thus, it is trustable. |
| 2 | Users' sensitive information needs to be secured and protected from external interferences. | Data in Transit Data at Rest Data in Process | Using a combination of a chain of signatures and blockchain receipt, we can easily detect if anyone has tampered with our database. Meanwhile, our blockchain-based secure channel can be used to provide secrecy during data transmissions. |
| 3 | The IoT system should assign a non-repudiable identification system to each of the IoT devices. | Data in Transit Data at Rest Data in Process | We design our identity system based on the public key mechanism, which provides a reliable non-repudiation guarantee. |
| 4 | Enterprises should maintain a metadata repository of the IoT devices for auditing purposes. | Data at Rest | The blockchain receipt can ensure the integrity of the metadata repository such that, when that metadata is shared with auditors, it still preserves the forensically sound guarantee. |
| 5 | The system may face difficulty in keeping IoT devices up to date. | Data in Process | The reputation system can force device owners to update their devices. For instance, we give weak ratings to outdated devices or have a policy that bans obsolete devices. |

**Table 8.** *Cont.*

| No | Open Problems | Related to | Our Solutions |
|----|--------------|-----------|---------------|
| 6 | Administrators also need to identify suspicious traffic patterns for incident management. | Data in Transit | While malicious traffic detections require other technologies (e.g., machine learning algorithm), we can assist this process by ensuring TLS communication integrity. For example, we can prove that a particular TLS handshake indeed happened. |
| 7 | The system also needs to have interoperability and protocol convergence to achieve efficient collaboration. | Data in Transit Data at Rest Data in Process | In our design, the blockchain serves as a standardized integrity platform for participants to collaborate in a secure and trustable manner. |
| 8 | The IoT data training generates low accuracy during the training of the analytic model. | Data in Process | While the actual training accuracy depends on the dataset, the model, and the algorithm, our platform forces participants to train honestly to achieve the best accuracy. |
| 9 | We need to have a parallel computation of the IoT data to support a multi-source platform. | Data in Process | The distributed training through split learning and federated learning come to the rescue. Our platform provides a complementary procedure to protect the integrity of those learnings. |

## 6.2. Future Considerations and Challenges

In this paper, we only propose the design and entrust implementations to interested adopters. Therefore, in this part, we discuss some future considerations and challenges.

**Public vs. private blockchain**: There are two types of blockchain: public (or permissionless) and private (or permissioned). The pioneer blockchain proposed in Bitcoin [4] is public, which allows anyone to join the blockchain network freely. Meanwhile, the private blockchain (e.g., Hyperledger Fabric [64]) requires authentication to join the network. Furthermore, the private blockchain enables private transactions [11], which allow the transaction to be encrypted such that only the sender and receiver can read it while others cannot. Adopters must understand the properties of the blockchain platform that they choose, whether it is public or private, and its implication to privacy. More specifically, since the data stored in the blockchain is visible to all nodes, they have to decide whether everyone is allowed to see the data or only authorized nodes need to know.

**Consensus-based scalability**: With the vast number of IoT data traffics, it is vital to assess our blockchain platform's scalability to cope with those high traffic demands. Since blockchain is a distributed system, its scalability depends on the underlying consensus algorithm. The trend is either to choose between the Proof-of-Work (PoW) or the Byzantine Fault Tolerance (BFT) algorithm [65]. Using PoW enables the blockchain to scale to thousands of nodes but suffers low throughput. On the other hand, BFT achieves higher throughput but fails to scale more than 16 nodes [66]. By knowing this issue, adopters have to carefully pick the blockchain platform that best suits their application, whether they want a high number of throughputs or a high number of nodes.

**Shift of trust**: the author in [67] defines trust as a probability that a person will perform a given action that is beneficial for the giver. The more likely that the person takes action means higher the trust. In centralized IoT, trust resides centrally in the IoT service. If the service can correctly perform IoT-related analytics, the trust will be higher for the company. We argue that using blockchain does not mean eliminating the trust system. Instead, it shifts the trust from the IoT service to the blockchain itself. In particular, the security of the blockchain relies heavily on the consensus operation. Therefore, we have to trust the blockchain network such that the notorious 51% attack or even the 25% attack [68] does not happen. If we use BFT as our consensus, then we need to make sure that not more than 33% of the nodes fail or become malicious at the same time [69]. Hence, adopters must understand this trust principle when developing the blockchain platform for IoT data integrity.

**Mitigation not remediation**: when a breach happens, security guarantees of blockchain are limited to preventing and detecting, not recovery. For example, in our example of the data at rest solution, we can easily detect attackers' tampering attempts using blockchain receipts. However, it is impossible to restore the data if our data is modified or, even worse, deleted during attacks. Therefore, adopters have to be aware of the importance of a backup system to remediate after attacks [70].

**Usability**: we should learn from PGP that poor usability due to complex key management and setup can result in people calling the system dead [71]. On the contrary, by observing the current trends, blockchain has better user experiences. First, blockchain has wallet applications to help users manage their private and public keys [72]. Second, developers create distributed applications (DApps) [73] as interfaces for users to interact easily with the blockchain platform. Third, the blockchain community is growing globally across many sectors [74]. This growing community will produce arguably more mature hardware and software components for future blockchain.

**Quantum computing**: finally, the blockchain's main threat is the rise of quantum computing that promises to break the underlying cryptography foundations such as hashing and RSA public-key cryptosystem [75]. Specifically, Grover's algorithm can dramatically speed up the hashing process to find collisions, easing modification of the blockchain's contents. Then, Shor's algorithm can factor large prime numbers faster, breaking the asymmetric-key cryptosystems' purpose. Therefore, further research regarding the future of the blockchain post-quantum era (e.g., proposed by Kiktenko et al. [76]) needs to start right now before the quantum technology arrives.

## 7. Conclusions

We proposed a grand design of blockchain-based continued integrity service for IoT big-data management in three IoT phases: data in transit, data at rest, and data in process. We first presented our motivations at each phase and surveyed related blockchain research from the literature as building blocks in constructing our design. Afterward, we laid out our solutions. For data in transit, we proposed decentralized identity management and secure channel establishment based on blockchain. For data at rest, we presented the use of a chain of signatures combined with blockchain receipts to augment the integrity of stored IoT data. We then designed the blockchain-based decentralized marketplace and federated learning for IoT entities to collaborate during data in process. As future works, interested adopters can try to implement our design in their IoT systems. With the building blocks already available, we argue that our proposal should be feasible to carry out. More importantly, a more in-depth exploration of the reputation and incentive mechanism should be the primary research directions. We argue that those two points are the heart of decentralization because they can force participants to obey the consensus rule.

**Author Contributions:** Conceptualization, Y.E.O. and S.-G.L.; methodology, Y.E.O.; validation, Y.E.O. and S.-G.L.; formal analysis, Y.E.O.; investigation, Y.E.O. and S.-G.L.; resources, Y.E.O. and S.-G.L.; writing—original draft preparation, Y.E.O.; writing—review and editing, Y.E.O., S.-G.L., and B.-G.L.; visualization, Y.E.O.; supervision, S.-G.L. and B.-G.L.; project administration, S.-G.L. and B.-G.L.; funding acquisition, S.-G.L. and B.-G.L. All authors have read and agreed to the published version of the manuscript.

**Funding:** This work was supported by Institute for Information and Communications Technology Promotion (IITP) grant funded by the Korea government (MSIT) (No. 2018-0-00245, Development of prevention technology against AI dysfunction induced by deception attack).

**Acknowledgments:** We would like to thank the anonymous reviewers for their comments and suggestions that helped us improve the paper.

**Conflicts of Interest:** The authors declare no conflict of interest.

**Abbreviations**

The following abbreviations are used in this manuscript:

| | |
|---|---|
| IoT | Internet of Things |
| TLS | Transport Layer Security |
| DTLS | Datagram Transport Layer Security |
| CoAP | Constrained Application Protocol |
| MQTT | Message Queuing Telemetry Transport |
| SLA | Service Level Agreement |
| API | Application Programming Interface |
| UUID | Universal Unique Identifier |
| PKI | Public Key Infrastructure |
| DNS | Domain Name Service |
| CA | Certificate Authority |
| PGP | Pretty Good Privacy |
| WoT | Web of Trust |
| ISP | Internet Service Provider |
| ECDSA | Elliptic Curve Digital Signature Algorithm |
| FL | Federated Learning |
| MEC | Mobile Edge Computing |
| IPFS | InterPlanetary File System |
| PoW | Proof-of-Work |
| BFT | Byzantine Fault Tolerance |
| DApp | Distributed Application |

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
