# Peer review of "Blockchain-Based Continued Integrity Service for IoT Big Data Management: A Comprehensive Design"

_electronics, doi:10.3390/electronics9091434_

Round 1
Reviewer 1 Report
Dear Authors,
Overall the idea of the paper is good. The content is also interesting for the readers. The paper however requires modifications:
- The paper needs to be proofed thoroughly for language- both grammar and tense.
- The methodology will need to be explained. Survey in the case of the paper is a literature survey and not a respondent survey. make this clearer. It is also not clear whether you have just proposed solutions or are trying these out experimentally.
- it will help to provide a flow diagram to understand the solutions specified in section 3- are these discrete, are these connected, do we need all of them to be implemented?
- More clarity is required between the research gap (proposed research questions or hypotheses) and the solutions, and this should be revisited when discussing the implications.
Author Response
First of all, we would like to thank you for your constructive feedback.
We appreciate all comments and suggestions for our paper.
1. The paper needs to be proofed thoroughly for language- both grammar and tense.
Because English is not our mother language, we are aware of our limitations. We have done our best to proof our paper thoroughly by manually rephrasing sentences, using grammar spell and checker programs, and consulting with our colleague.
2. The methodology will need to be explained. Survey in the case of the paper is a literature survey and not a respondent survey. make this clearer. It is also not clear whether you have just proposed solutions or are trying these out experimentally.
The survey that we did is a literature survey. We have revised our writings to specifically mentioned the word "literature survey." We also briefly explain our methodology in Section 1 (before the contributions' paragraph).
In this paper, we only propose several protocol designs for IoT integrity service, and we omit implementations for future works.
We stated this remark in the beginning paragraph of Section 6.2.
3. it will help to provide a flow diagram to understand the solutions specified in section 3- are these discrete, are these connected, do we need all of them to be implemented?
Yes, they are connected, and we need all of them. We have added a flow diagram in Section 3 for each proposal: decentralized identity management and secure channel establishment. Furthermore, we also remodel other flow diagrams in Section 4 and 5 to explain our proposed design better.
4. More clarity is required between the research gap (proposed research questions or hypotheses) and the solutions, and this should be revisited when discussing the implications.
We edited the subsections' structure in Section 3, 4, and 5 from "Issues and Challenges" and "Blockchain Solutions" to "Motivations," "Our Proposed Solutions," and "Quick Analysis."
We hope that these changes can solve the mentioned issues.
Because we introduce a privacy-preserving IoT data flow pipeline, we think we must add a federated learning process based on blockchain as complementary solutions for Data in Process. You can find this addition in Section 5.2.2.
Finally, you can download the attached PDF, which shows our editing with the tracked changes features enabled. We generate the PDF using the latexdiff package. Note that, for some reason, the latexdiff does not detect some modifications we made in the PDF. For example, the title, abstract, figure, and reference differences are not annotated. Nevertheless, we hope that the PDF can help to show the granularity of our revision.

Reviewer 2 Report
The paper analyses blockchain solutions for some relevant problems, which are emerging In the provision of data integrity services for IoT big data management and proposes the design of an integrity protocol covering three phases of IoT operations, namely the transmission of IoT data, the storage of the data in the database and the data processing.
The paper is interesting and well fitting the aims and scope of the journal. The treated topic is of relevant interest for the scientific and technical community.
The introduction correctly frames the proposed work in the international context and duly highlight the contribution of the proposed approach. However, the main elements of novelty of the proposed approach should be better summarized at this stage. As a minor further remark, I suggest the authors to remove the subtitles “contribution” and “roadmap”, which are not necessary and to present such topics in a more discursive way.
The authors show a relevant knowledge of the state of the art, which is correctly analysed in order to describe the main aspects of the proposed approach.
The proposed description of the IoT Big Data Management is very clear and comprehensive.
In Section 3 an interesting survey of Blockchain Solutions for Data in Transit is proposed. However, many symbols are used without a clear introduction of their meaning. This is a relevant weakness, that should be overcome in order to make the paper suitable to publication.
Similar considerations hold for Sections 4 and 5.
A further weakness is related to the fact that the proposed design of the blockchain integrity platform is not clearly identifiable and needs to be extrapolated by the reader. The authors should make an effort in order to separate the analysis of the state of the arte and the description of their solution, which should be proposed before the discussion of its benefits with respect to existing solutions.
The paper would greatly benefit from the addition of a list of used acronyms and symbols, which would improve the paper readability.
As a formal remark, the English language is not totally incorrect, but it is somehow rough and many repetition and colloquial expressions are present, which should be removed as they are not suitable for a scientific paper. I suggest the authors to exploit the support of a professional proofreader.
Author Response
First of all, we would like to thank you for your constructive feedback.
We appreciate all comments and suggestions for our paper.
The introduction correctly frames the proposed work in the international context and duly highlight the contribution of the proposed approach. However, the main elements of novelty of the proposed approach should be better summarized at this stage. As a minor further remark, I suggest the authors to remove the subtitles “contribution” and “roadmap”, which are not necessary and to present such topics in a more discursive way.
We omit the "contribution" and "roadmap" words from the paragraph.
We also briefly explain our methodology in Section 1 (before the contributions' paragraph).
In Section 3 an interesting survey of Blockchain Solutions for Data in Transit is proposed. However, many symbols are used without a clear introduction of their meaning. This is a relevant weakness, that should be overcome in order to make the paper suitable to publication.
Similar considerations hold for Sections 4 and 5.
We have added a list of used notations in Table 3, 5, and 7.
A further weakness is related to the fact that the proposed design of the blockchain integrity platform is not clearly identifiable and needs to be extrapolated by the reader. The authors should make an effort in order to separate the analysis of the state of the arte and the description of their solution, which should be proposed before the discussion of its benefits with respect to existing solutions.
We edited the subsections' structure in Section 3, 4, and 5 from "Issues and Challenges" and "Blockchain Solutions" to "Motivations," "Our Proposed Solutions," and "Quick Analysis."
We hope that these changes can solve the mentioned issues.
The paper would greatly benefit from the addition of a list of used acronyms and symbols, which would improve the paper readability.
We have added a list of used notations in Table 3, 5, and 7. We also put a list of acronyms used in this paper in the Abbreviations section.
As a formal remark, the English language is not totally incorrect, but it is somehow rough and many repetition and colloquial expressions are present, which should be removed as they are not suitable for a scientific paper. I suggest the authors to exploit the support of a professional proofreader.
We have done our best to proof our paper thoroughly by manually rephrasing sentences, using grammar spell and checker programs, and consulting with our colleague.
Because we introduce a privacy-preserving IoT data flow pipeline, we think we must add a federated learning process based on blockchain as complementary solutions for Data in Process. You can find this addition in Section 5.2.2.
Finally, you can download the attached PDF, which shows our editing with the tracked changes features enabled. We generate the PDF using the latexdiff package. Note that, for some reason, the latexdiff does not detect some modifications we made in the PDF. For example, the title, abstract, figure, and reference differences are not annotated. Nevertheless, we hope that the PDF can help to show the granularity of our revision.

Round 2
Reviewer 1 Report
I am happy with the revised version of the paper. The paper has new ideas and it will benefit researchers in the field of Blockchain and IoT systems.
Author Response
I am happy with the revised version of the paper. The paper has new ideas and it will benefit researchers in the field of Blockchain and IoT systems.
We want to thank you for your constructive feedback.
We appreciate all comments and suggestions for our paper.
The new revised version has several minor grammar and sentence updates.
We also fix minor errors in Figure 3.
The detail of our revisions can be seen in the attached PDF.

Reviewer 2 Report
The authors significantly improved the paper according to the suggestions provided by the reviewers. The quality of the paper is improved and now it is suitable to publication.
Author Response
The authors significantly improved the paper according to the suggestions provided by the reviewers. The quality of the paper is improved and now it is suitable to publication.
We want to thank you for your constructive feedback.
We appreciate all comments and suggestions for our paper.
The new revised version has several minor grammar and sentence updates.
We also fix minor errors in Figure 3.
The detail of our revisions can be seen in the attached PDF.
